# All that structure matches does not glitter

**Maya M. Martirossyan[1], Thomas Egg[1], Philipp Höllmer[1],**
**George Karypis[2], Mark Transtrum[3], Adrian Roitberg[4], Mingjie Liu[4],**
**Richard G. Hennig[4], Ellad B. Tadmor[2], Stefano Martiniani[1] [*]**

[1]New York University [2]University of Minnesota [3]Brigham Young University [4]University of Florida

## Abstract

Generative models for materials, especially inorganic crystals, hold potential to transform the theoretical prediction of novel compounds and structures. Advancement in this field depends critically on robust benchmarks and minimal, information-rich datasets that enable meaningful model evaluation. This paper critically examines common datasets and reported metrics for a crystal structure prediction task—generating the most likely structures given the chemical composition of a material. We focus on three key issues: First, materials datasets should contain unique crystal structures; for example, we show that the widely-utilized *carbon*-24 dataset only contains $\approx 40\%$ unique structures. Second, materials datasets should not be split randomly if polymorphs of many different compositions are numerous, which we find to be the case for the *perov*-5 and *MP*-20 datasets. Third, benchmarks can mislead if used uncritically, e.g., reporting a match rate metric without considering the structural variety exhibited by identical building blocks. To address these oft-overlooked issues, we introduce several fixes. We provide revised versions of the *carbon*-24 dataset: one with duplicates removed, one deduplicated and split by number of atoms $N$, one with enantiomorphs, and two containing only identical structures but with different unit cells. We also propose new splits for datasets with polymorphs, ensuring that polymorphs are grouped within each split subset, setting a more sensible standard for benchmarking model performance. Finally, we present *METRe* and *cRMSE*, new model evaluation metrics that can correct existing issues with the match rate metric.

## 1 Introduction

Recent advances in machine learning (ML) have fueled enormous interest in its application to materials science. For instance, machine-learning interatomic potentials have enabled efficient molecular simulations at near density-functional theory (DFT)-level accuracy [1, 2]. ML has also been applied to experiment planning and reaction prediction, enabling autonomous decision making in the laboratory through planning agents [3, 4]. This work concerns generative models for inorganic crystal structures, which learn mappings from a tractable base distribution to novel structures and compositions resembling the training data. This field has recently gained momentum, with numerous frameworks and architectures regularly claiming state-of-the-art performance [5–21].

The availability of high-quality and diverse datasets is paramount in the training and benchmarking of generative models. Minimal test datasets provide fast feedback during the development of generative models, prior to expensive training on large datasets. The bulk of materials datasets for the explicit purpose of materials discovery are generated using random structure searches with DFT [22, 23]. However, the influence of polymorphs (*i.e.*, different crystal structures for the same chemical compound) and structural duplicates in such standard datasets for inorganic crystal generation (see Fig. 1a–d), especially in the smallest test datasets, has largely been overlooked.

---

[*]Corresponding author: sm7683@nyu.edu.

39th Conference on Neural Information Processing Systems (NeurIPS 2025) Track on Datasets and Benchmarks.

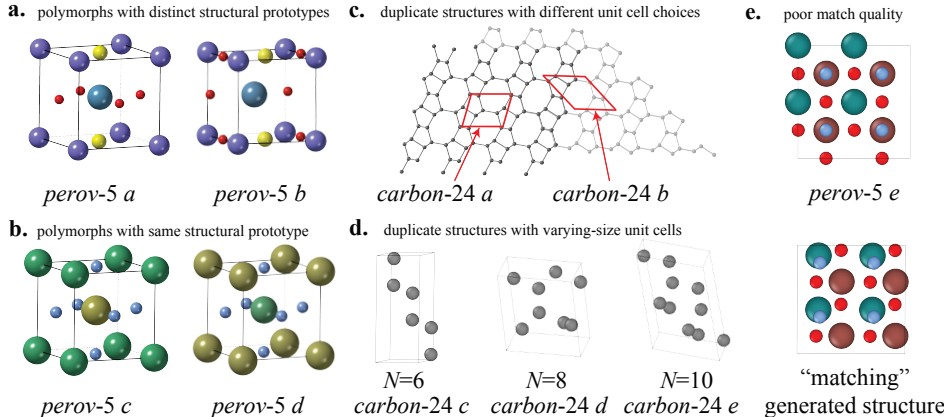

**a.** polymorphs with distinct structural prototypes  **c.** duplicate structures with different unit cell choices  **e.** poor match quality

*perov*-5 *a*    *perov*-5 *b*       *carbon*-24 *a*    *carbon*-24 *b*       *perov*-5 *e*

**b.** polymorphs with same structural prototype  **d.** duplicate structures with varying-size unit cells

*perov*-5 *c*    *perov*-5 *d*       $N$=6              $N$=8            $N$=10            "matching"
                                      *carbon*-24 *c*  *carbon*-24 *d*  *carbon*-24 *e*   generated structure

Figure 1: Enumerating existing features of datasets and benchmarks used in crystal structure prediction for generative models of inorganic crystals. (**a**) Two *perov*-5 structures of composition $CaCdSO_2$, but with different structural prototypes in which structure $b$ is a distorted version of structure $a$. (**b**) Two *perov*-5 structures of composition $HfNbN_3$, with the same structural prototype but with the elements at the A and B sites (Hf and Nb) swapped in the perovskite $ABX_3$ structural prototype. (**c**) Two *carbon*-24 duplicate structures (one in dark and the other in light gray) with their unit cells marked in red. (**d**) Three *carbon*-24 duplicate structures with different unit cell sizes. (**e**) Views along a lattice direction of (top) a *perov*-5 test set structure and (bottom) a structure from a generative model which are considered "matching" despite significant structural distortions between the two, calculated using Pymatgen's `StructureMatcher` module with standard tolerances `ltol`$= 0.3$, `stol`$= 0.5$, `angle_tol`$= 10.0$.

In addition to the datasets, the benchmark metrics themselves must be adequate to validate the quality of the generated samples and, therefore, to judge and compare different generative models [24, 25]. For the crystal-structure prediction (CSP) task—in which a generative model attempts to generate the positions and lattice vectors for a given composition—the match-rate metric is well established and thus reported in most works [5, 7–9, 12–16, 21, 26–28]. As we will discuss, however, the structure-matching procedure underlying this metric has limitations that must be overcome (see Fig. 1e).

In our paper, we demonstrate several examples where datasets and benchmarks have not been generated with the underlying scientific questions in mind. We elucidate the presence of a significant fraction of duplicate structures in the *carbon*-24 dataset, the presence of polymorphic pairs of crystals with the same composition but different structure split randomly across the *perov*-5 dataset(s), and benchmarking with match rates which lose meaning in the presence of polymorphs. We propose solutions through the publication of new datasets and dataset splits in addition to new benchmarks for assessing CSP task performance. We provide a brief crystallography primer with background on crystal-structure representations in Appendix F.

## 2 Related work

### 2.1 Crystal structure prediction and polymorphism

Crystal structure prediction aims to predict stable phases from a given composition. Polymorphs are distinct crystalline phases for the same chemical composition and are plentiful in the realm of experimental structural synthesis. Famously, inorganic compounds such as calcium carbonate can nucleate and grow in the aragonite, calcite, and vaterite crystalline phases [29]. Other well-known cases include carbon and its many allotropes—such as diamond, graphene, graphite, and buckminsterfullerene (buckyballs)—as well as silicon—which at both ambient condition and under pressure forms a large number of crystal phases [30, 31]. For molecular crystals, polymorphism is already well-understood to be the chief difficulty for CSP due to small free energy differences between stable polymorphs [32–34] which are pertinent to synthesis and drug design [35]. Even non-crystalline systems such as metamorphic proteins can adopt different stable, folded structures [36].

Thus, structure prediction from composition in generative models should thus consider the propensity to form various possible structural phases *from the same building blocks*. Although the standard datasets for CSP of atomic crystals contain polymorphs (as, *e.g.*, by design in the *carbon*-24 dataset of carbon structures), their influence on performance metrics was previously not studied explicitly.

## 2.2 Existing datasets

In the literature, generative CSP models have been trained on very few datasets which have become the standard in the materials science domain. This paper is mainly concerned with three of them. The ***carbon*-24** dataset contains 10 153 structures consisting purely of carbon and containing up to 24 atoms in the unit cell[1] [15]. It was curated from a ten-times larger dataset of carbon structures obtained at a pressure of 10 GPa in an *ab initio* random structure search [37] by choosing the structures with the lowest energy per atom. The ***perov*-5** dataset contains 18 928 perovskite structures [38]. Here, each unit cell contains five atoms with varying cell sizes (all cubic in shape) and chemical compositions. The ***MP*-20** dataset contains 45 229 structures from the Materials Project with up to 20 atoms per unit cell spanning a diverse range of unit cell shapes and compositions [15, 39].

The comparatively small *carbon*-24 and *perov*-5 datasets could, in principle, serve as minimal datasets with low computational cost during training and benchmarking. However, as we will discuss in Section 3, they contain duplicate structures and polymorphs that may result in misleading performance metrics. The *MP*-20 dataset does not suffer as severely from these problems. Thoughtful benchmarks for *de novo* generation (DNG) from models trained on *MP*-20 [40] are actively being expanded, while benchmarks for crystal structure prediction lag behind—even though good CSP models can be utilized for DNG if provided with novel compositions [5, 41].

## 2.3 Existing metrics

Benchmarking generative models for inorganic crystal structure prediction involves generating a structure for every composition in a test set. A typically reported metric is the **match rate** computed using Pymatgen's `StructureMatcher` module [42] which performs a one-to-one comparison between the generated and reference structure. Here, the structures have to "match" only to some tolerance determined by the `stol`, `ltol`, and `angle_tol` parameters of the `StructureMatcher`: `stol` restricts how great the discrepancy between two sets of atomic sites can be, normalized by the average free length per atom $\sqrt[3]{V/N}$ where $V$ is the volume of the (matched) unit cell and $N$ is the number of atoms; `ltol` defines the fraction by which unit cell lengths are allowed to differ; `angle_tol` provides a bound on the difference in angle between matched unit cell vectors [42]. The alignment of two approximately matching structures is computed by an algorithm which reduces structures to their primitive cells, aligns lattice vectors within `ltol` tolerance, changes the basis of lattice vectors from one structure's to the other's—giving access to the (normalized) root-mean square error between the atom positions between two structures. This typically reported metric is the mean **RMSE**, that is, the per-particle average root-mean-square error between matched generated and test structures. Non-matching structures are ignored for the computation of the mean RMSE.

For the *carbon*-24 dataset that consists entirely of different structures of the same composition, the match-rate metric is naturally ill-defined because of its one-to-many nature [5, 9, 12]. Some works alternatively report a $k$-match rate [8, 9, 12, 14, 28], where $k = 20$ structures are generated for every given composition in the test set. If at least one of the $k$ generated structures matches the reference structure, the lowest-RMSE match is counted—thus $k$ match rate considers possible polymorphs of crystals of the same composition in a statistical manner. If the generative model is able to generate several stable polymorphs (as desired), only one of the $k$ trials has to yield a structure matching the specific structure in the test set in order to obtain a high $k$-match rate. However, evaluation of the $k$-match rate comes at a significantly higher computational cost, and $k$ would need to be scaled with the expected number of polymorphs in the training data. An additional discussion of the $k$-match rate in comparison to the proposed metric of this paper is provided in Appendix A.1.

Thermodynamic (meta-)stability of generated structures (*i.e.*, having a negative or small energy above the convex hull of known stable structures) is an established metric for the *de novo* generation task of generative models for inorganic crystals [5, 6, 9–12, 20], where the model predicts both structure and composition. However, this is not a feasible metric for the *carbon*-24 and *perov*-5 datasets that

---

[1]A unit cell is a periodic building block that tiles space to form a crystalline material (see Appendix F).

include metastable structures by design [15, 37, 38]. For example, diamond is expected to be the only thermodynamically stable structure in the *carbon*-24 dataset.

## 2.4 Generative Models

In this work, we evaluate the performance of three generative models on various versions of the datasets introduced in Section 3. They perform either diffusion modeling [43, 44] or flow-based generative modeling [45, 46]—two major generative modeling paradigms. The first model, **DiffCSP** [14], is an equivariant diffusion model while the second one, **FlowMM** [12], is a flow-based generative model that applies the conditional flow matching framework [47]. The last model, **OMatG** [5], is a flow-based generative model which implements a general stochastic interpolant framework encompassing both diffusion modeling and conditional flow matching as special cases [45, 48].

## 3 Datasets

In this section, we discuss the deduplication (see Section 3.1) and polymorph-aware splitting (see Sections 3.2 and 3.3) of established datasets. We stress that no structural edits are made to individual crystals in this process: structures are either removed entirely, or regrouped into new splits. All new datasets can be found online at: `https://huggingface.co/collections/colabfit/datasets-all-that-structure-matches-does-not-glitter`.

## 3.1 Carbon structures

We show that the *carbon*-24 dataset contains far fewer unique structures than previously understood. An identification method for duplicates built upon Pymatgen's `StructureMatcher` reveals that less than half of the 10 153 structures published in the dataset are, in fact, distinct. Consequently, we introduce two new variants: ***carbon*-24-unique** (see Section 3.1.1), which treats enantiomorphs[2] as duplicates, ***carbon*-24-unique-with-enantiomorphs** (see Section 3.1.2), which retains enantiomorphs as distinct structures, and the related toy dataset ***carbon*-enantiomorphs** with only the chiral pairs. The single-element nature of this data allows us to design additional benchmark datasets. We introduce the ***carbon*-24-unique-$N$-split** datasets (see Section 3.1.3), which make it possible to systematically study how well generative models can extrapolate beyond their training data to different unit cell sizes $N$. Finally, we explicitly use the identified duplicate structures to generate the ***carbon*-X** and ***carbon*-NXL** datasets for "overfitting" tests (see Section 3.1.4). We provide links to all of these datasets in Appendix B.

We note that our proposed identification method for duplicates based on the `StructureMatcher` can only provide highly *likely* duplicate candidates because it is still based on a limited numerical evaluation. Even defining a structure "match" is inherently ambiguous: Different settings can change whether two structures are considered matching or distinct. In dataset creation for generative models, we argue that the tolerance thresholds we set are sensible and informative given the current limits of CSP model performance.

### 3.1.1 Pruning duplicates

Pymatgen's `StructureMatcher` has a variable tolerance for the comparison of two structures. The tolerances for the match-rate computation in the CSP task of generative models are generally chosen quite large (`stol`= 0.5, `ltol`= 0.3, and `angle_tol`= 10.0 which, in fact, exceed the default values of `stol`= 0.3, `ltol`= 0.2, and `angle_tol`= 5.0) [5, 7–9, 12–16, 21, 26–28]. Such loose tolerances may be reasonable when comparing imperfect structures obtained from generative models, which necessarily come with some uncertainty relative to the "perfect" crystals in the reference dataset, though their impact should still be carefully assessed (see Section 4.2). We note, however, that these tolerances are unsuitable for evaluating the structural distinctness within the *carbon*-24 dataset itself.

In order to reasonably compare structures within *carbon*-24, we dynamically vary the tolerances of the `StructureMatcher`. For every pair of structures in the dataset, we find the match-boundary values of the `stol`, `ltol`, and `angle_tol` parameters where two structures transition from matching to non-matching. We use separate binary searches for every parameter while keeping the other ones

---

[2]Structures that are mirror images of each other but cannot be superimposed through translation or rotation.

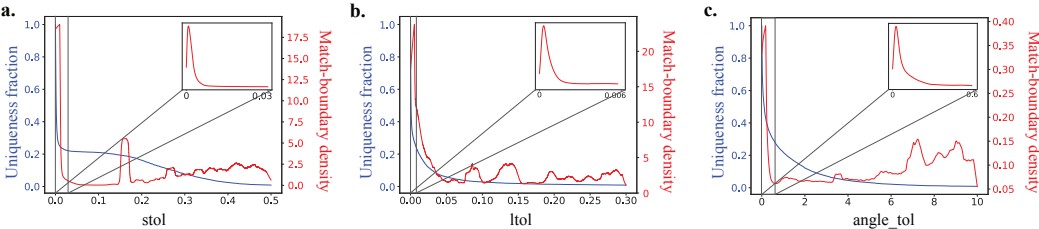

Figure 2: Kernel density estimates (with tophat kernel for large plots and Gaussian kernel for insets) of the distributions of match-boundary tolerance and uniqueness fraction for (**a**) `stol`, (**b**) `ltol`, and (**c**) `angle_tol` performed on the *carbon*-24 dataset. These densities only count structure pairs which are considered matching at or below the maximum tolerances, and ignore structure pairs which are too structurally distinct to match.

fixed at their loose values `stol`= $0.5$, `ltol`= $0.3$, and `angle_tol`= $10.0$. The `stol` parameter is not utilized in the alignment process; we make the simplifying approximation that the `ltol` and `angle_tol` tolerances can be treated independently. Further details on these computations is provided in Appendix J.

We show the distributions of the match-boundary tolerances for every tolerance parameter in Fig. 2. They all show signatures of a large peak at very low tolerance which is a clear sign of duplicate structures in the dataset. This is also confirmed by the estimated fraction of unique structures as a function of the tolerances in Fig. 2. This fraction reaches $1.0$ only at very small values of the tolerance parameters. We conclude that the structure pairs within the peak at low tolerances represent replicated crystal structures that were not previously identified. The unit cells in the dataset can thus only be deemed all "distinct" if symmetries that leave the crystal structure unchanged are ignored. Unit cells, however, are *non-unique* representations of crystal structures, and an infinite number of choices of repeating units can be made which tile space to produce the crystal structure of interest (see Fig. 1c and d).

From Fig. 2, we estimate threshold values for each tolerance parameter below which the large peaks, indicative of duplicates structures, appear (`stol`= $0.025$, `ltol`= $0.002$, and `angle_tol`= $0.4$). Using these thresholds, we generated three lists of duplicated structures (one for every tolerance parameter) that we combine into a single list by retaining only the pairs that appear in all three of them. After grouping the pairs into clusters, treating duplicates as mutual, we create a novel *carbon*-24-unique dataset by selecting a single representative from each cluster. This conservative cut leaves $4250$ structures (down from $10\,153$) from which we create training, validation, and test sets with a random 60–20–20 % split.

### 3.1.2 Enantiomorph pairs

Certain chiral structures form enantiomorph pairs, mirror images that cannot be superimposed by any combination of proper rotations or translation.[3] We noticed that chiral enantiomorph pairs were being tagged as duplicate structures by Pymatgen's `StructureMatcher` since it allows for improper rotations (such as mirrors or inversions) in order to map two structures to one another. To identify enantiomorph pairs we disabled improper rotation mappings in `StructureMatcher` and recomputed the RMSE for all previously identified duplicate pairs. Pairs exhibiting a tenfold or greater increase in RMSE under this constraint were reclassified as enantiomorphs rather than duplicates.

We release the *carbon*-24-unique-with-enantiomorphs dataset which retains both structures in each enantiomorph pair and explicitly labels them. We found $80$ enantiomorph pairs; we note that this screening was only applied to the structures in the *carbon*-24-unique dataset. We benchmark performance of models trained on one of each chiral pair in *carbon*-enantiomorphs (see H.3).

### 3.1.3 Datasets split by $N$

The single-element nature of the *carbon*-24-unique dataset provides a unique opportunity to isolate the effect of increasing size and structural complexity with the number of carbon atoms $N$. We thus

---

[3]A real-world example of a chiral pair of objects are human hands.

introduce *carbon*-24-unique-$N$-split datasets, comprising non-random splits of the *carbon*-24-unique dataset that are organized by $N$. Structures are grouped into training, validation, and test sets by increasing (low-to-high) or decreasing (high-to-low) $N$, aiming for as close to a 60–20–20 % split as allowed by the groupings of $N$. For the low-to-high split, the training set contains 2280 structures with $N = 6$–10 atoms, the validation set contains 1159 structures with $N = 12$–14, and the test set contains 811 structures with $N = 16$–24. Vice versa, for the high-to-low split, the training set contains 2633 structures with $N = 10$–24, the validation set contains 792 structures with $N = 8$, and the test set contains 825 structures with $N = 6$.

Organizing the data by $N$ allows us to systematically study how generative models generalize across different scales. This is also consequential for dataset creation, as smaller unit cells are significantly less expensive to obtain with DFT. Beyond carbon, such extrapolation is essential for modeling realistic materials systems that exhibit chemical or structural disorder, large unit cells, or even molecular motifs as in molecular crystals.

### 3.1.4 Datasets of duplicates

Pruning the *carbon*-24 dataset of duplicates provides the opportunity to create datasets in which all crystals are identical to one another but with different choices of unit cells. From identified duplicate pairs, we publish and benchmark the use of two such datasets for use in "overfitting" tests for generative models. The first is the *carbon-X* dataset, which contains 480 carbon duplicate structures which have the same number of atoms $N$ and cell shape $L$ but different translations of the fractional coordinates $X$. The second is the *carbon-NXL* dataset, which contains 353 carbon duplicate structures that have different numbers of atoms per unit cell ($N = 6$–16), different cell shapes $L$ and fractional coordinates $X$ (see Fig. 1c and d). As these two datasets each contain only a single structure and can be used to test whether the model can generate that singular structure, the datasets are not split.

These duplicate datasets are special because they are augmented with respect to an important type of symmetry—the equivalence of different unit cell choices for the same crystal—which standard encoders such as CSPNet [14] are not equivariant with respect to. CSPNet and the MatterGen model encoder [6] break invariance to this symmetry by injecting information about the lattice vectors or angles into their graph representations.

## 3.2 Polymorph-aware splits for perovskite structures

Unlike the *carbon*-24 dataset, the *perov*-5 dataset does not contain duplicates. It does, however, contain 9282 polymorph pairs (totaling 18 564 structures) and only 364 compositions that show up once in the dataset. These pairs are structurally dissimilar with either structural distortions (as in Fig. 1a) or elements swapped (as in Fig. 1b).

The full dataset was randomly split in a 60–20–20 % fashion by Xie et al. [15] into training, validation, and test sets, which raises the question: How are the structures in each polymorph pair distributed? There are 2265 composition matches between the validation and training set (out of 3787 validation set structures) and 2214 composition matches between the test and training set (out of 3785 test set structures). Only 94 validation set structures and 107 test set structures are considered "matching" with high RMSEs of $\approx 0.4$–0.5 to those in the training set, confirming high structural dissimilarity between the composition-matched structures. The random split of polymorph pairs into training, validation, and test sets implies that generative models are trained on one set of structures—and subsequently evaluated on their ability to generate a different structure of the same composition. We argue that this is a poor benchmark: even with a perfect model, it would be highly improbable that the precise structure in the test set be the one that is generated.

We publish and benchmark new splits for the *perov*-5 dataset that we call ***perov*-5-polymorph-split**, which confine polymorph pairs to be in the same portion of the split. For the evaluation over the validation and test sets, generative models will thus have to attempt to generate both structurally distinct structures of entirely unseen compositions. Under the assumption that a refined match-rate metric can handle polymorphs (see Section 4.1), this is arguably both a more reasonable task—with expectations for out-of-distribution generation adjusted—but also a harder task—generating multiple structures per composition for entirely new compounds—for benchmarking.

### 3.3 Polymorph-aware splits for large, diverse datasets

The *MP*-20 dataset also contains polymorphs: $37\,217$ unique reduced compositions across $45\,229$ total structures ($\sim 82\%$ unique compositions). In contrast to the *perov*-5 dataset, however, the fraction of non-unique compositions is much smaller. We provide new polymorph-aware splits *MP*-20 dataset, termed ***MP*-20-polymorph-split**. Unlike for the resplitting of the *perov*-5 dataset, we consider how the propensity for a given composition to exhibit polymorphism could exhibit dependence on the number of unique elements of the material (commonly termed $n$-arity). In creating new splits for *MP*-20, polymorphs of the same composition were assigned to the same split, and the polymorphs groups were distributed such that the distribution of the $n$-arity of the combined dataset matched that of each individual split.

## 4 Benchmarking CSP model performance

### 4.1 Amending benchmarks to be robust to polymorphs

Datasets with many polymorphs, like the *carbon*-24 and *perov*-5 datasets, break the typically reported match-rate metric. Even if a generative model could produce all polymorphs of a given composition, it would score poorly because match-rate evaluates each generated structure against only one reference structure with the same composition. This one-to-one approach forces models to "learn" a unique structure per composition, ignoring the true multiplicity of (meta-)stable polymorphs and introducing an incorrect physical assumption.

We introduce the *match everyone to reference* (**METRe**) metric—pronounced ˈmēt-ər, like the SI unit—to assess how well generated structures cover the test set. Unlike standard match rate, METRe compares every reference structure against every generated ("match everyone") and counts a match whenever a generated structure falls within tolerance of the reference structure ("to reference"), selecting only the best match per reference when computing the RMSE, as shown in Fig. 3a–e. The METRe rate is then the fraction of reference structures that find at least one match.

Counting "matches to everyone" with respect to generated structures is counterproductive because a model could have a high-scoring match metric by generating structures that resemble only a small fraction of reference structures. By contrast, METRe "matches to everyone" with respect to reference structures and does not have this issue. For datasets with many polymorphs (such as *carbon*-24), the ability to reproduce this structural diversity is essential, and METRe naturally accounts for it and rewards this behavior by counting matches with respect to the reference (test) set. In the limit of no polymorphism, the METRe rate reduces to the original definition of the match rate. In addition to the METRe metric, the mean RMSE and cRMSE (introduced in Section 4.2) between every reference structure and the best matching generated structure, as shown in Fig. 3, is equally if not more important. We provide Python code for the computation of the METRe and the cRMSE scores in Appendix A.

We emphasize that the $k$-match rate (see Section 2.3) is fundamentally different from the METRe rate as the latter is measuring matches with respect to the entire test set. In future work, one could consider an analogous $k$-METRe rate where the generated set is larger than the reference set thus mitigating the effect of statistical fluctuations in the generation of different polymorph structures. We add as a final note that METRe becomes inflated and harder to interpret correctly if there are many duplicates in the test set—which is undesirable in the context of generative modeling—and therefore duplicate structures should be removed from the dataset before using the METRe rate.

### 4.2 New metric to combine RMSE and match rate

Optimizing generative models only with respect to match or METRe rates, say in a hyperparameter sweep, may lead to models that poorly match to a large number of the test set structures (see Fig. 1e where two structures with little structural similarity are considered matching). The application of `StructureMatcher` to compute structure matches is highly tolerant—for example, usage with standard tolerances to compute matches would suggest that the uniqueness rate within the *carbon*-24 dataset (see Fig. 2) is between 3–4 %. Thus, METRe alone is not a sufficiently strong metric for optimizing generative models for crystalline materials. Similarly, the mean RMSE metric alone is also insufficiently qualitative because the RMSE between two structures is only computed if structures are matched. In the worst case, models may learn to generate only a single structure from

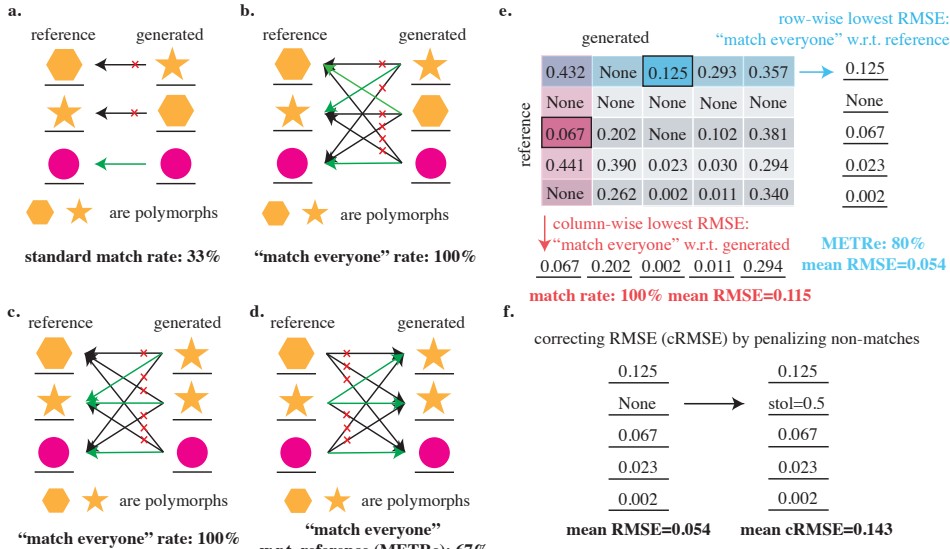

Figure 3: Demonstrating prior and new benchmarks. (**a–d**) A toy-case, in which the same colored shapes are considered polymorphs, shows different ways of computing match rate: (**a**) standard match rate, which penalizes polymorphs in the generated set being out of order; (**b**) "match everyone" metric, which fixes the fictitious penalty in (a); (**c**) a case of the "match everyone" metric in which a high match rate can be achieved without generating the diversity of polymorph structures; (**d**) our solution to the problems posed in (a) and (c), in which the number of matches from the "match everyone" metric is counted with respect to the reference set. (**e**) A demonstration of how "match everyone" differs when computed with respect to the generated vs. reference structures, showing that only the metric with respect to the reference structures (METRe) catches cases in which none of the generated structures match a given reference structure. (**f**) The implementation of corrected RMSE on a given matching metric.

the test set to high accuracy. Compatible with this discussion, we note that the recent work on the OMatG model observed an apparent tradeoff between the match-rate metric and the mean RMSE [5].

We propose a new corrected RMSE (**cRMSE**) metric that combines the METRe and RMSE metrics, as illustrated in Fig. 3e. We define cRMSE by penalizing non-matching structures by using `stol` as the non-matching RMSE (instead of ignoring the missing match). We choose `stol` as the penalty because it sets the threshold for the computed RMSE of the aligned structures in `StructureMatcher` (if a mapping can be found).

We note that the cRMSE metric can also be defined with the original definition of the match-rate metric. It is a general way to combine any match-rate metric with an RMSE for the optimization of generative models. We also emphasize that mean cRMSE can be rewritten as a combination of any type of match rate and corresponding mean RMSE as a function of the `stol` used with `StructureMatcher`. We propose that the **primary benchmark for CSP performance should be the mean cRMSE**(`stol`) instead of the match or METRe rate and RMSE separately.

## 5 Results

We benchmark DiffCSP, FlowMM, and OMatG on our new datasets using METRe and cRMSE, with cRMSE as the primary performance metric, using the standard `stol`= 0.5, `ltol`= 0.3, `angle_tol`= 10.0 for `StructureMatcher`. This means that all reported cRMSE values are a function of `stol` = 0.5. Hyperparameter choices (using published ones for DiffCSP and FlowMM) and optimization (hyperparameter tuning for lower cRMSE for OMatG) are discussed in Appendix D. The flexibility of OMatG allows us to study a wide variety of models that are differentiated by the choice of a positional interpolant (for more details, see Ref. [5]). We further note that all of the standard match rates and METRe results are reported without any filtering for structural or

Table 1: Benchmarking generative models (OMatG labeled by positional interpolant) on the new *carbon*-24-unique and *perov*-5-polymorph-split datasets, as well as the original *perov*-5 datasets using the proposed METRe match rate, mean RMSE, and corrected mean cRMSE metrics. For the *carbon*-24-unique generated structures, we also report the result of standard match rate and corresponding RMSE for comparison.*

| Model | *carbon*-24-unique | | *perov*-5 | *perov*-5-polymorph-split |
|---|---|---|---|---|
| | Std. Match % (↑) / RMSE (↓) | METRe % (↑) / RMSE (↓) / **cRMSE** (↓) | METRe % (↑) / RMSE (↓) / **cRMSE** (↓) | METRe % (↑) / RMSE (↓) / **cRMSE** (↓) |
| DiffCSP* | **21.2** % / 0.380 | 98.2 % / 0.231 / 0.235 | 57.7 % / **0.072** / **0.253** | **78.9** % / 0.072 / 0.162 |
| FlowMM* | 19.5 % / 0.358 | 98.4 % / 0.193 / 0.198 | 58.4 % / 0.096 / 0.264 | 78.8 % / 0.070 / 0.161 |
| OMatG-LinearODE | 19.8 % / 0.286 | 98.0 % / 0.183 / 0.189 | 67.5 % / 0.236 / 0.322 | 76.8 % / 0.055 / **0.158** |
| OMatG-LinearODE$\gamma$ | 16.9 % / 0.314 | 97.6 % / 0.213 / 0.220 | 76.3 % / 0.344 / 0.381 | 75.9 % / 0.067 / 0.172 |
| OMatG-TrigODE | 18.8 % / **0.272** | **98.5** % / 0.183 / **0.187** | **84.3** % / 0.359 / 0.381 | 77.1 % / 0.059 / 0.160 |
| OMatG-TrigODE$\gamma$ | 19.8 % / 0.307 | 98.1 % / **0.181** / **0.187** | 75.7 % / 0.313 / 0.358 | 76.3 % / **0.053** / 0.159 |
| OMatG-EncDecODE | 18.1 % / 0.298 | 98.2 % / 0.195 / 0.201 | 72.6 % / 0.398 / 0.425 | 74.5 % / 0.058 / 0.171 |
| OMatG-SBDODE | 14.8 % / 0.324 | 97.8 % / 0.218 / 0.224 | 85.1 % / 0.366 / 0.386 | 77.1 % / 0.062 / 0.163 |

*Starred model names have identical hyperparameters for both *perov*-5 splits. OMatG models were hyperparameter tuned for maximizing standard match rate on *perov*-5 and minimizing cRMSE on *perov*-5-polymorph-split.

compositional validity (as in Ref. [5]). The filtering is not necessary as high RMSE or cRMSE values will indicate poor quality of matches with greater propensity for structural invalidity. We also report our new benchmarks on old datasets: for *perov*-5 (see Table 1) and *MP*-20 (see Table 2).

In Table 1, we compare the performance of the models on the *carbon*-24-unique and *perov*-5-polymorph-split datasets. We also include results for the original *perov*-5 dataset split for comparison. For the *carbon*-24-unique dataset, we measure the performance on identical generated structures with both standard match (one-to-one) and METRe rates and highlight the significant jump in fraction of matches identified by accounting for polymorphism. Comparing the RMSE values between standard matching and METRe, we also note a $\approx 0.1$ decrease in the average RMSE for matching structures. Finally, for METRe we also compute the cRMSE, which is close to the RMSE values since the METRe value is high. Overall for the *carbon*-24-unique dataset, the METRe rate and its corresponding RMSE and cRMSE values indicate the strongest performance for trigonometric positional interpolants using OMatG, followed closely by the performance for linear flow-matching with both OMatG and FlowMM.

For the *perov*-5-polymorph-split dataset, we assess the models' performances using METRe, RMSE and cRMSE, and compare the results to those obtained for models trained on the *perov*-5 split. Arguably, the *perov*-5-polymorph-split is a challenging objective because the model is expected to produce two structures (recall that each composition admits two polymorphs in this dataset) from compositions that it has never encountered. Nevertheless, the model performance on the *perov*-5-polymorph-split dataset is improved relative to the previous *perov*-5 dataset across most METRe rates and all METRe-associated RMSE and cRMSE values. Again, the strongest performance in terms of RMSE and cRMSE is obtained for linear and trigonometric interpolant OMatG models, while the strongest performance for METRe was for DiffCSP; differences in cRMSE, however, are modest between all models. These results suggest that by simply splitting the *perov*-5 data differently, the models are better able to generalize not only to new compositions but also to new structural prototypes.

For the *MP*-20-polymorph-split dataset, we include in Table 2 results for the METRe, RMSE, and cRMSE metrics for models trained on the previous (*MP*-20) and the new (*MP*-20-polymorph-split) dataset splits. Structures for the DiffCSP and FlowMM models were generated using published *MP*-20 hyperparameters. For DiffCSP and FlowMM, performance on the polymorph-aware dataset split declined in comparison to the original dataset split. This is unsurprising given that the hyperparameters were tuned without polymorph-aware benchmarks on the original dataset split. For OMatG models— through a hyperparameter optimization procedure for both dataset splits—we observed a modest improvement in performance and higher state-of-the-art performance metrics.

We also benchmark on the "duplicates" datasets and show results in Table 3 for *carbon-NXL* and in Table 4 in Appendix H.1 for *carbon-X*. For these datasets, we restrict benchmarks to only the OMatG conditional flow-matching model (OMatG-Linear) and compare results for the standard CSPNet encoder to an augmented CSPNet—which adds both lattice angle information as well as the number of atoms $N$ to the representation. We report the standard match rate for these benchmarks, because the test set (*i.e.*, the training set) contains only a single crystal structure. For the *carbon-NXL* dataset, we additionally benchmark the models by isolating reported metrics by $N$, pinpointing the difficulty of generating identical structures with more atoms. These datasets provide idealized conditions in

Table 2: Benchmarking generative models on the *MP*-20 and *MP*-20-polymorph-split datasets using the proposed METRe match rate, mean RMSE, and corrected mean cRMSE metrics. DiffCSP and FlowMM models both use published *MP*-20 hyperparameters (consistent across the two datasets, signified by the $*$ next to the model name). The OMatG model was hyperparameter tuned to optimize for high match rate on *MP*-20 and low cRMSE on the *MP*-20-polymorph-split dataset.

| Model | *MP*-20 | | | *MP*-20-polymorph-split | | |
|---|---|---|---|---|---|---|
| | METRe % ($\uparrow$) | RMSE ($\downarrow$) | cRMSE ($\downarrow$) | METRe % ($\uparrow$) | RMSE ($\downarrow$) | cRMSE ($\downarrow$) |
| DiffCSP* | 58.8 % | 0.064 | 0.244 | 53.1 % | 0.084 | 0.279 |
| FlowMM* | **67.0** % | 0.067 | 0.210 | 65.2 % | 0.079 | 0.226 |
| OMatG-LinearODE | 66.0 % | **0.058** | **0.208** | **70.5 %** | **0.056** | **0.187** |

Table 3: Benchmarking the *carbon-NXL* duplicates dataset using mean RMSE, corrected mean cRMSE, and standard match rate (chosen because there is only one unique structure in the dataset). Training and generation initialization were both performed with the entire dataset. Results are reported for the complete dataset and broken down by unit cell size $N$. A conditional flow-matching OMatG-LinearODE model was used with two choices of encoders, CSPNet and augmented CSPNet with lattice angle and $N$ information. We exclude metrics for $N = 10$–16 due to deficiency of such structures in both the train and test dataset and, thus, the unpredictability of the generated structures.

| Model | *carbon-NXL* | | | | | | | | |
|---|---|---|---|---|---|---|---|---|---|
| | All $N$ | | | $N = 6$ | | | $N = 8$ | | |
| | Std. Match (%) $\uparrow$ | RMSE $\downarrow$ | **cRMSE** $\downarrow$ | Std. Match (%) $\uparrow$ | RMSE $\downarrow$ | **cRMSE** $\downarrow$ | Std. Match (%) $\uparrow$ | RMSE $\downarrow$ | **cRMSE** $\downarrow$ |
| CSPNet | 47.3 % | 0.008 | 0.266 | 60.0 % | 0.005 | 0.203 | 39.0 % | 0.013 | 0.310 |
| aug-CSPNet | 47.7 % | 0.006 | 0.264 | 69.2 % | 0.005 | 0.157 | 26.0 % | 0.010 | 0.373 |

which no compositional complexity and exactly one structural prototype needs to be learned by the model, and difficulty of the task can be controlled systematically by varying $N$.

Performance deteriorates for the *carbon-NXL* dataset as the number of atoms $N$ and lattice vectors $L$ change, with only 60–69% match rate for structures with $N = 6$ and significantly lower match rates of 26–39% for $N = 8$, along with RMSE values an order of magnitude higher (Table 3). To our knowledge, this is the first study for inorganic crystals to provably demonstrate that performance is limited not just by structural or compositional complexity, but also by the dimensionality of the learned flows as defined by the unit-cell size $N$.

To further examine the impact of $N$, we use the hyperparameters from models trained on the *carbon*-24-unique dataset and report METRe, RMSE, and cRMSE in Table 5 in Appendix H.1 for models trained on the *carbon*-24-unique-$N$-split datasets. Comparing the low-to-high to the high-to-low $N$-split, we find that the latter yields significantly better results. This is to be expected: we already demonstrated that low-$N$ structures are considerably better at achieving high-fidelity matches. The low-to-high $N$-split performs poorly and serves as a challenging objective for future generative models to target.

# 6   Discussion

We have shown that progress demands not only advanced generative models but also meticulously curated, task-aligned datasets and evaluation metrics designed for the specific challenges within crystal structure prediction. By systematically analyzing widely-used benchmarks for CSP, we uncover ill-posed assessments and improperly curated datasets. To rectify these issues, we introduced new curated datasets and dataset splits and benchmarks that expand the scope of evaluating CSP performance. Our results demonstrate that improved dataset design and evaluation criteria lead to better performance on more difficult tasks. Our analysis also revealed that the performance of generative models degrades with unit-cell size $N$, elucidating a clear challenge for generative models. We hope that our datasets, metrics and benchmarks will contribute to the foundation of this field, encouraging more rigorous practices in model evaluation and dataset design.

## Acknowledgments and Disclosure of Funding

The authors would like to thank Shenglong Wang at NYU IT HPC and Gregory Wolfe for their support in this work. The authors acknowledge funding from NSF Grant OAC-2311632. P. H. and S. M. also acknowledge support from the Simons Center for Computational Physical Chemistry (Simons Foundation grant 839534, MT). The authors gratefully acknowledge the computational resources and consultation support that have contributed to the research results reported in this publication, provided by: IT High Performance Computing at New York University; the Empire AI Consortium; UFIT Research Computing and the NVIDIA AI Technology Center at the University of Florida in part through the AI and Complex Computational Research Award.

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

# A METRe and cRMSE metrics

For a mathematical definition of the mean cRMSE metric, let $N_{\text{test}}$ be the number of test set structures, $N_{\text{ref.match}}$ the number of matches according to the METRe metric, and $\text{RMSE}_i$ the relevant RMSE for the $i$th structure in the test set. We can then express the mean cRMSE as

$$\text{mean cRMSE}(\texttt{stol}) = \frac{\sum_{i=1}^{N_{\text{ref.match}}} \text{RMSE}_i + \texttt{stol}(N_{\text{test}} - N_{\text{ref.match}})}{N_{\text{test}}} \tag{1}$$

$$= \text{METRe} * (\text{mean RMSE} - \texttt{stol}) + \texttt{stol},$$

where we used $\text{METRe} = N_{\text{ref.match}}/N_{\text{test}}$ and $\text{mean RMSE} = \sum_{i=1}^{N_{\text{ref.match}}} \text{RMSE}_i / N_{\text{ref.match}}$.

In the following, we provide Python code that computes the METRe metric, the mean RMSE and the mean cRMSE metrics based on a list of generated structures, a list of test set structures and tolerance parameters for Pymatgen's `StructureMatcher`.

The code for calculation of the METRe metric and mean cRMSE is available within the OMatG software hosted at `https://github.com/FERMat-ML/OMatG`.

## A.1 Comparison to $k$-match rate

As discussed in Section 2.3, the $k$-match rate depends on a fixed integer value for $k$ that determines the number of generated structure for each reference structure. If one of these $k$ generated structures matches the reference structure, a match is recorded. This approach ameliorates the problem of polymorphism since the model has more opportunities to generate the correct polymorph under consideration. By scaling $k$, the probability of producing the reference polymorph increases. In order to assess whether the generative model is able to generate all possible polymorphs, $k$ should be at least the number of maximum polymorphs in the dataset.

METRe possesses two key advantages over the $k$-match rate. First, METRe requires no explicit definition of $k$. Instead, METRe only requires as many generated structures as are present in the test dataset. As a result, METRe is more efficient at inference time. Second, METRe wastes no structure in that each generated structure is compared against each reference structure. It automatically rewards the generation of polymorphs in proportion to their appearance in the dataset. In this light, METRe can be thought of as a $k$-match rate where $k$ is inferred from the number of polymorphs of each chemical composition in the test dataset. The efficiency gains come from the fact that none of the structures generated in the computation of METRe are discarded.

One potential downside of METRe with respect to $k$-match rate is the number of calls to PyMatGen's `StructureMatcher` which are necessary. Since each reference structure and each generated structure must be compared, METRe has a worst-case time complexity of $\mathcal{O}(n^2)$. However, this is rarely the case as only materials with matching chemical stoichiometry can be matched. Structures with incongruent stoichiometry are automatically rejected making the algorithm computationally efficient in most practical settings.

## B  Data availability

The original *carbon*-24 and *perov*-5 datasets were released under the MIT license in the GitHub repository of CDVAE [15]: `https://github.com/txie-93`. We also release polymorph-split versions of *MP*-20 [39] and Alex-*MP*-20 [6, 49, 50] keeping polymorphs in the same split and ensuring that the distribution of $n$-arity—the number of unique elements $n$ in each crystal structure— did not change between the combined dataset and each of the splits.

All datasets introduced in this work are released under the CC-BY 4.0 license on Huggingface under the following links:

- *carbon*-**24-unique** and *carbon*-**24-unique-**$N$**-split** — Dataset of unique carbon structures derived from the original *carbon*-24 dataset treating enantiomorph pairs as duplicates: `https://huggingface.co/datasets/colabfit/carbon-24_unique`

- *carbon*-**24-unique-with-enantiomorphs** — Dataset of unique carbon structures derived from the original *carbon*-24 dataset treating enantiomorph pairs as distinct: `https://huggingface.co/datasets/colabfit/carbon-24_unique_with_enantiomorphs`

- *carbon*-**enantiomorphs** — Only the 80 chiral pairs from *carbon*-24-unique-with-enantiomorphs, split into a train and validation set with opposite-handedness structures index-matched: `https://huggingface.co/datasets/colabfit/carbon-enantiomorphs`

- *carbon*-*X* — A dataset of one particular carbon crystal structure with fixed number of atoms $N = 6$ and lattice vectors $L$ but under various translations of fractional coordinates $X$: `https://huggingface.co/datasets/colabfit/carbon_X`

- *carbon*-*NXL* — A dataset of one particular carbon crystal structure with different unit-cell representations that vary all $N$, $X$, and $L$: `https://huggingface.co/datasets/colabfit/carbon_NXL`

- *perov*-**5-polymorph-split** — New splits for the *perov*-5 dataset which restrict polymorph pairs to be in the same part of each split: `https://huggingface.co/datasets/colabfit/perov-5_polymorph_split`

- *MP*-**20-polymorph-split** — New splits for the *MP*-20 dataset which restrict polymorph pairs to be in the same part of each split: `https://huggingface.co/datasets/colabfit/MP-20-polymorph-split`

- **Alex-***MP*-**20-polymorph-split** — New splits for the Alex-*MP*-20 dataset which restrict polymorph pairs to be in the same part of each split: `https://huggingface.co/datasets/colabfit/Alex-MP-20_Polymorph_Split`

## C  Code availability

Pymatgen and its `StructureMatcher` are released under the MIT license: `https://github.com/materialsproject/pymatgen`

We additionally list links and licenses to the different open-source generative models that we evaluated in this work:

- DiffCSP [14] is released under the MIT license: `https://github.com/jiaor17/DiffCSP`

- FlowMM [12] is released under the CC-BY-NC license: `https://github.com/facebookresearch/flowmm`

- OMatG [5] is released under the MIT license: `https://github.com/FERMat-ML/OMatG`

## D  Hyperparameter choices

Hyperparameter selection is crucial to the performance of the three generative models that we investigated in this work. For FlowMM and DiffCSP, we chose the hyperparameters from these works which yielded the best performance for both the *carbon*-24 and *perov*-5 datasets [12, 14]. For OMatG, we performed hyperparameter optimization to minimize the cRMSE metric using the `Ray`

`Tune` package [51] along with the HyperOpt Bayesian optimization library [52]. For more details on the hyperparameter search spaces, see Höllmer et al. [5].

OMatG models discussed throughout this work are labeled by the interpolating function used to learn the fractional coordinates $X$. For more details on the functional forms of these interpolants, we refer to Albergo et al. [45] and Höllmer et al. [5].

Model checkpoints and accompanying hyperparameters for the OMatG models trained in this study will be accessible at: `https://huggingface.co/OMatG`

## E   Cost of training and optimization

Here we report the cost of model training for DiffCSP, FlowMM, and OMatG as well as hyperparameter optimization for OMatG [5, 12, 14]. For training on both *carbon*-24-unique-$N$-split datasets, we trained DiffCSP, FlowMM, and two versions of OMatG (standard and augmented) for 8000 epochs on either NVIDIA RTX8000, V100 or A100 GPUs. For training OMatG on *carbon-X* and *carbon-NXL* we trained one version with a standard CSPNet encoder and one with an augmented CSPNet encoder which breaks invariance to unit cell choice for 8000 epochs for each dataset on either NVIDIA RTX8000 or V100 GPUs.

For hyperparameter optimization of each different OMatG version on the *carbon*-24-unique, *perov*-5, and *perov*-5-polymorph-split datasets we trained on 2 NVIDIA A100 GPUs for 5 days for each model. For training DiffCSP and FlowMM on these three datasets we used NVIDIA A100 GPUs each for 8000, 6000, and 6000 epochs respectively.

# F  Crystallography primer

Crystallography deals with the study and classification of crystal structures. Idealized crystal structures are infinite point patterns which contain long-range translational periodic order.[4]

**Lattices**  The translational symmetry of crystals is captured by their crystal *lattices* (also called Bravais lattices). There are five in two dimensions and fourteen in three dimensions.

A lattice can be described by the discrete set of points generated by integer linear combinations of a set of linearly independent basis vectors:

$$\Lambda = \{\mathbf{R} = n_1\mathbf{a}_1 + n_2\mathbf{a}_2 + n_3\mathbf{a}_3 \mid n_i \in \mathbb{Z}\}.$$

The vectors $\mathbf{a}_1, \mathbf{a}_2, \mathbf{a}_3$ span the repeating unit of the lattice, called the *unit cell*, whose volume is given by

$$V_{\text{cell}} = |\mathbf{a}_1 \cdot (\mathbf{a}_2 \times \mathbf{a}_3)|.$$

Lattices should not be conflated with structures: for example, the commonly-referred to *fcc* (face-centered-cubic) is a lattice, and not a crystal structure; the crystal structure is termed monatomic *fcc* (sometimes termed cubic-close-packed, or *ccp*), where a particle sits directly on each lattice point. More generally, a crystal structure can be viewed as the combination of a Bravais lattice and a *basis* of particles attached to each lattice point:

$$\Lambda + \{\mathbf{r}_\alpha\}_{\alpha=1}^{N_{\text{basis}}},$$

where each $\mathbf{r}_\alpha$ specifies the position of an atom within the unit cell.

**Space group symmetry**  In three dimensions, *space groups* combine translations with rotations, inversions/reflections, screw axes, and glide planes. The crystal space-group symmetry partitions space into sets of symmetry-equivalent points called Wyckoff positions. Each Wyckoff position is characterized by a site-symmetry group—the subgroup of the space group that leaves a representative point of that position fixed—and particles occupy one or more of these positions. Wyckoff positions can either be general (for arbitrary coordinates $(x, y, z)$) or special (possessing higher site-symmetry and reduced free parameters compared to the general position). For each space group, there are an infinite number of possible crystal structures. Crystals are classified by the full (maximal)symmetry space group of the structure, but they may also be represented by subgroups of their space group). For example, space group $P1$ has one Wyckoff position with free parameters $(x, y, z)$—any crystal structure can be classified as space group $P1$, though this classification is not useful if the structure possesses higher symmetry.

Space group tables are available through the International Union of Crystallography (IUCr) [53]. Standard notations include Hermann–Mauguin (international), Schoenflies, and Hall symbols.

**Unit cells**  Crystal structures can be defined by their *unit cell*, composed of the lattice vectors, the particle coordinates, and the chemical identities of the particles. A fully specified unit cell generates a unique periodic crystal structure, though a structure has many equivalent unit-cell representations. Degenerate representations can be related by unimodular transformations of the lattice vectors:

$$\mathbf{a}_i' = \sum_j M_{ij}\mathbf{a}_j, \qquad M \in GL(3, \mathbb{Z}), \quad \det M = 1.$$

These equations can be summarized as requiring volume preservation (through the allowed values of the determinant) and redefining the basis vectors by integer combinations of one another. We assume the number of particles in the cell is constant, therefore excluding supercells—which are yet another way to generate unit cells with different lattice vectors. In summary, the same physical crystal can be represented by infinitely many equivalent unit cells.

There are two types of standardized unit cells: conventional and primitive. The primitive cell is the smallest volume that, when translated by all lattice vectors $\mathbf{R} \in \Lambda$, fills all of space without overlaps or gaps, and contains exactly one lattice point.[5] Primitive cells are not uniquely defined; however, the

---

[4]We forego discussion of quasiperiodic order for the purposes of this primer.

[5]One lattice point corresponds to the number of particles that are in the basis.

Niggli-reduced cell can be computed algorithmically [54] and is a unique choice of the primitive cell. Although the primitive cell is the minimal repeating unit of a crystal, the conventional cell is often preferred in crystallography because it makes the underlying symmetry of the lattice and crystal more explicit. Conventional cells are defined differently for each crystal system (*e.g.*, cubic, tetragonal, orthorhombic) to highlight characteristic symmetry axes and planes. This process of symmetrizing a unit cell from its primitive to conventional cell can be performed using the *spglib* software [55]. Although not standardized, *supercells* can be generated by replicating unit cells along lattice vectors to create larger periodic volumes—for example, to model defects or finite-size effects.

We walk through these concepts in a crystallography primer notebook with the *carbon-NXL* dataset, which is accessible at `https://www.kaggle.com/code/mayamartirossyan/crystal-representations-primer`.

# G   Tolerance sensitivity analysis of cRMSE and METRe

We include results in Fig. 4 for benchmarking the tolerance sensitivity of both the METRe and cRMSE metrics on the changing of `ltol`, `stol`, and `angle_tol`. Our results suggest a sensitivity to tolerance based on the match-quality of the generated structures, which is inferred from their performance metrics (METRe and cRMSE). Generated datasets with higher-quality matches are less sensitive to the `StructureMatcher` tolerance parameters. Moreover, across the board it is clear that `stol` has the most impact in determining both METRe rate and cRMSE.

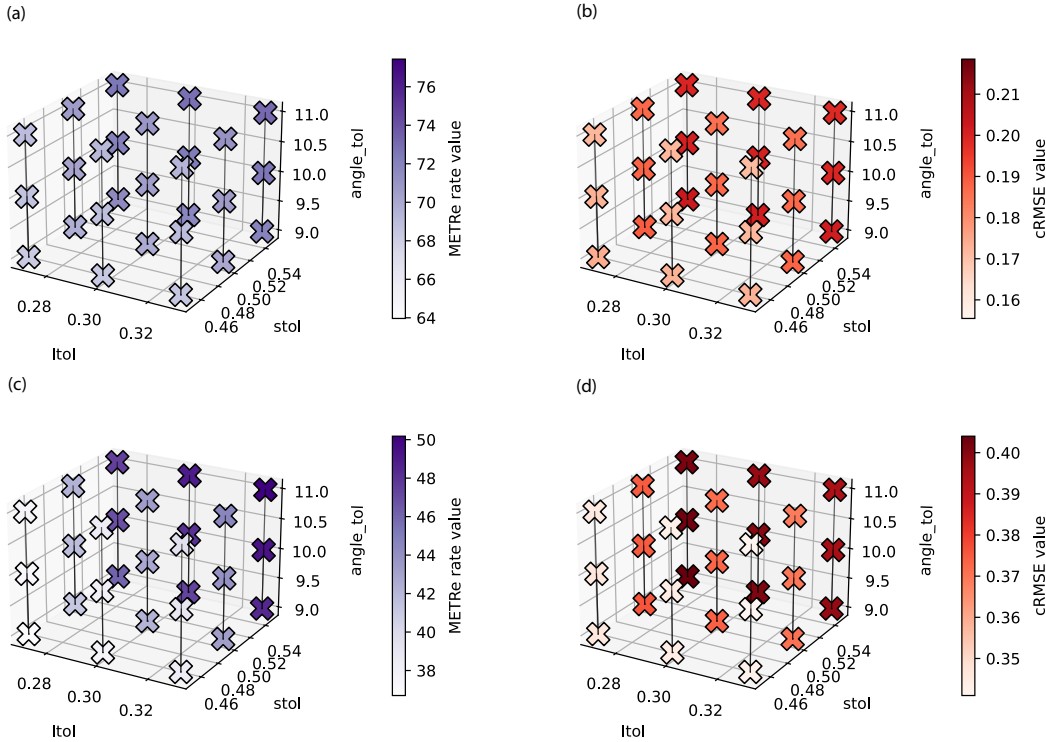

Figure 4: Tolerance sensitivity plots for OMatG models trained on the polymorph-split *MP*-20 dataset. **(a–b)** Best-performing model with linear positional interpolant and ODE sampling and **(c–d)** worst-performing model with trigonometric interpolant with the latent variable $\gamma$ and ODE sampling. METRe rates are shown for (a) and (c) and cRMSE values are shown for (b) and (d); color bars have equivalently-sized ranges across subfigures. Vertical lines are drawn for clarity.

# H  Additional discussion and evaluations on *carbon*-24-derived datasets

## H.1  Evaluations on *carbon-X* and *carbon*-24-unique-$N$-split

We include below benchmarking results for *carbon-X* (Tab. 4) and the two *carbon*-24-unique-$N$-split dataset splits (Tab. 5).

The *carbon-X* match rate is 100% (Table 4), which is unsurprising given that both CSPNet and the OMatG model—which explicitly corrects for the system's center-of-mass to make flows—are translation invariant. For *carbon*-24-unique-$N$-split, the split with low-$N$ structures in the training set, combined with high-$N$ structures in the test set, results in poor performance compared to both the high-to-low split as well as the *carbon*-24-unique dataset.

Table 4: Benchmarking *carbon-X* with mean RMSE, corrected mean cRMSE, and standard match rate because the dataset contains one unique crystal. The OMatG-LinearODE framework is used with two choices of encoders.

| Model | *carbon-X* | | |
|---|---|---|---|
| | Std. Match (%) ↑ | RMSE ↓ | **cRMSE** ↓ |
| CSPNet | 100.0 % | 0.001 | 0.001 |
| aug-CSPNet | 100.0 % | 0.001 | 0.001 |

Table 5: Benchmarking performance of generative models DiffCSP, FlowMM, and OMatG-LinearODE on the *carbon*-24-unique-$N$-split datasets with both increasing (low-to-high) and decreasing (high-to-low) atoms per unit cell $N$. Match rate and RMSEs are computed with the METRe metric.

| Model | *carbon*-24-unique-$N$-split (low → high) | | | *carbon*-24-unique-$N$-split (high → low) | | |
|---|---|---|---|---|---|---|
| | METRe (%) ↑ | RMSE ↓ | **cRMSE** ↓ | METRe (%) ↑ | RMSE ↓ | **cRMSE** ↓ |
| DiffCSP | 96.7 % | 0.426 | 0.429 | 100.0 % | 0.077 | 0.077 |
| FlowMM | 97.4 % | 0.404 | 0.406 | 100.0 % | 0.043 | 0.043 |
| OMatG | 96.3 % | 0.398 | 0.402 | 100.0 % | 0.045 | 0.045 |

## H.2  Re-evaluating *carbon*-24-unique

We perform re-evaluation on the *carbon*-24-unique dataset both in the context of uniqueness (Fig. 6).

In Fig. 5, we plot the distribution of energies per atom both before and after pruning of the *carbon*-24 dataset.

In Fig. 6, we repeat the analysis of Section 3.1.1 on the *carbon*-24-unique dataset that was pruned of identified duplicates. As in Fig. 2 for the original *carbon*-24 dataset, we show the distributions of the match-boundary tolerances for every tolerance parameter. In the deduplicated dataset, the large peak at low parameter values of `angle_tol` is entirely gone. While there are still visible peaks for the other two parameters, these peaks are now shifted to higher tolerance values. Also, the increase of the estimated fraction of unique structure towards zero tolerances is less pronounced. The remaining presence of the peaks in the distributions for `stol` and `ltol` can be explained by our conservative determination of duplicates, where two structures must be considered close with respect to all three tolerance parameters to be counted as a duplicate.

We do not directly compare generative models trained on datasets with duplicates and those trained on deduplicated datasets: Our reasoning for this is that both cases require a unified choice of a metric of sample quality and a shared test dataset. In our case, *carbon*-24 and *carbon*-24-unique do not meet either criteria. For instance, consider benchmarking with METRe—which is the appropriate choice given the 'polymorphism' the carbon-only datasets exhibit, but is unsuitable to datasets with duplicate structures. Therefore, it would not provide a useful benchmark on the *carbon*-24 dataset. Conversely, using the choice of standard match rate as a test metric is not informative because when polymorphs are present a one-to-one matching algorithm is not sensible.

While one could benchmark a model trained on a training set with duplicates on a deduplicated test dataset, care must be taken to ensure that there is no data leakage between the duplicated training

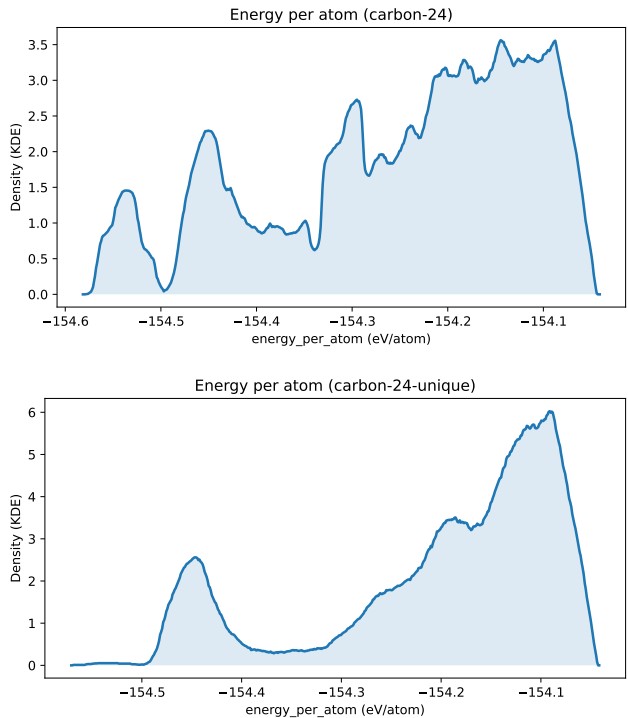

Figure 5: Kernel density estimates (with tophat kernel) of the distributions of energies per atom for **(a)** the *carbon*-24 dataset and **(b)** the *carbon*-24-unique dataset presented in this work.

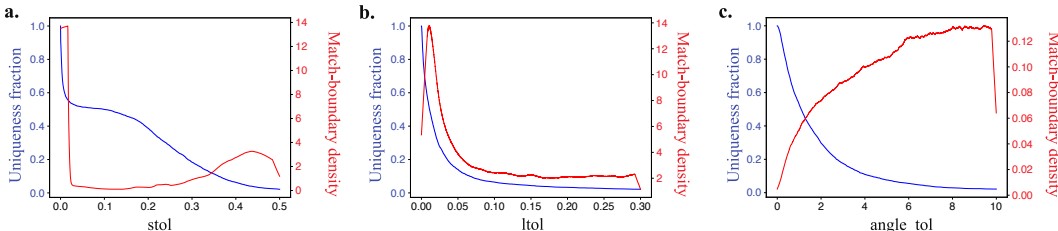

Figure 6: Tophat kernel density estimate of the distributions of match-boundary tolerance and uniqueness fraction for **(a)** `stol`, **(b)** `ltol`, and **(c)** `angle_tol` performed on the *carbon*-24-unique dataset. These densities only count structure pairs which are considered matching at or below the maximum tolerances, and ignore structure pairs which are too structurally distinct to match.

set and the test dataset. In this work, we deduplicated the *carbon*-24 dataset, and randomly split this into training, validation, and test datasets. Therefore, we did not ensure that the deduplicated test dataset and the original training set with duplicates do not have any crystal structures in common. Creating a shared test set is the clearest way to avoid data leakage, but is more straightforward to implement if one is augmenting a (deduplicated) dataset with duplicate structures; it is significantly more challenging to implement if de-duplication is required from a dataset containing duplicates, such as in our case.

We emphasize that datasets containing duplicates, including the original datasets, should be used where the prevalence of duplicate crystal structures is useful for the task at hand: for example, if one is attempting data augmentation for different unit cell representations. Therefore, training on duplicate structures is a completely reasonable objective as long as the presence of duplicate structures is documented and known.

### H.3 Enantiomorphs

We benchmark the ability for our model to produce structures of different handedness by benchmarking model performance on the toy-dataset *carbon*-enantiomorphs, which is composed of only chiral carbon structures. The dataset is split into a training and validation set such that structures of opposite handedness are not in the same split. An OMatG model with linear positional interpolant was trained for 4000 epochs on the *carbon*-enantiomorphs training dataset, and the best validation loss checkpoint was used for generation. The model is then expected to generate either structures of the same or opposite handedness, and we investigate this by comparing METRe rate using two versions of `StructureMatcher`—one as-is and another which has improper rotations disabled. The latter is done by enabling the output of the rotation matrix found by the lattice mapping, and subsequently requiring that its determinant be positive.

Results are shown in Tab. 6. If the model memorized the training set structures, it would show poor performance with the inversion-disabled `StructureMatcher` and better performance with the standard `StructureMatcher` for comparisons between the generated structures and the validation set. In this case, the model was not especially successful at predicting even the correct structures, evident in poorer performance in comparison to the baseline set by the benchmarks comparing the training and validation structures with the inversion-disabled `StructureMatcher`. We note, however, that the similarity in the results between the generated structures *vs.* validation set and the generated structures *vs.* training set for both `StructureMatcher`s. The results suggest a very modest preference for handedness learned and poor performance across the board.

Table 6: Six comparisons of structures measured with METRe, RMSE, and cRMSE. Both the standard implementation of `StructureMatcher` as well as an inversion-disabled version of it are utilized. The comparisons of the training and validation sets serve as a baseline for interpreting results.

|  | METRe (%) ↑ | RMSE ↓ | **cRMSE** ↓ |
|---|---|---|---|
| Standard `StructureMatcher` | | | |
| Generated structures *vs.* validation set | 90.0% | 0.347 | 0.362 |
| Generated structures *vs.* training set | 90.0% | 0.348 | 0.363 |
| Training set *vs.* validation set | 100.0% | 0.003 | 0.003 |
| Inversion-disabled `StructureMatcher` | | | |
| Generated structures *vs.* validation set | 90.0% | 0.358 | 0.372 |
| Generated structures *vs.* training set | 90.0% | 0.356 | 0.371 |
| Training set *vs.* validation set | 100.0% | 0.217 | 0.217 |

# I Quantifying uncertainty for benchmarks

Below we provide standard error values from multiple generation runs with different seeds for *carbon*-24-unique (Table 7), *perov*-5-polymorph-split (Table 8), both *carbon*-24-unique-$N$-split low-to-high and high-to-low (Table 9), *carbon-NXL* (Table 10), and *carbon-X* (Table 11). We also include results for a modification of *carbon-X* in Table 12, in which six additional unit cells of the same crystal structure but with $N = 12$ carbon atoms are added during training to the existing $479$ structures, but generation results are presented only for $N = 6$ atoms; we note the order of magnitude worse performance compared to Table 11. We use three generation runs as done *via* Miller et al. [12] for all tables excepting Table 10, 11, and 12.

Table 7: Standard errors for three generation runs from the same checkpoints reported for METRe, RMSE, and cRMSE values for the *carbon*-24-unique dataset.

| Method | *carbon*-24-unique | | |
|---|---|---|---|
| | METRe % ($\uparrow$) | RMSE ($\downarrow$) | cRMSE ($\downarrow$) |
| DiffCSP | $98.0 \pm 0.2\,\%$ | $0.229 \pm 0.001$ | $0.234 \pm 0.001$ |
| FlowMM | $98.1 \pm 0.1\,\%$ | $0.1930 \pm 0.0003$ | $0.199 \pm 0.001$ |
| OMatG-LinearODE | $97.6 \pm 0.2\,\%$ | $0.181 \pm 0.001$ | $0.189 \pm 0.001$ |
| OMatG-LinearODE$\gamma$ | $97.7 \pm 0.3\,\%$ | $0.212 \pm 0.001$ | $0.219 \pm 0.001$ |
| OMatG-TrigODE | $98.3 \pm 0.1\,\%$ | $0.1825 \pm 0.0005$ | $0.1880 \pm 0.0005$ |
| OMatG-TrigODE$\gamma$ | $98.1 \pm 0.1\,\%$ | $0.180 \pm 0.001$ | $0.187 \pm 0.002$ |
| OMatG-EncDecODE | $98.5 \pm 0.1\,\%$ | $0.200 \pm 0.003$ | $0.205 \pm 0.002$ |
| OMatG-SBDODE | $97.8 \pm 0.1\,\%$ | $0.218 \pm 0.001$ | $0.225 \pm 0.001$ |

Table 8: Standard errors for three generation runs from the same checkpoints reported for METRe, RMSE, and cRMSE values for the *perov*-5-polymorph-split dataset.

| Method | *perov*-5-polymorph-split | | |
|---|---|---|---|
| | METRe % ($\uparrow$) | RMSE ($\downarrow$) | cRMSE ($\downarrow$) |
| DiffCSP | $77.4 \pm 0.8\,\%$ | $0.069 \pm 0.001$ | $0.166 \pm 0.003$ |
| FlowMM | $78.2 \pm 0.3\,\%$ | $0.071 \pm 0.001$ | $0.165 \pm 0.002$ |
| OMatG-LinearODE | $76.8 \pm 0.1\,\%$ | $0.0555 \pm 0.0003$ | $0.1593 \pm 0.0005$ |
| OMatG-LinearODE$\gamma$ | $75.9 \pm 0.1\,\%$ | $0.0670 \pm 0.0002$ | $0.1713 \pm 0.0003$ |
| OMatG-TrigODE | $76.7 \pm 0.3\,\%$ | $0.0586 \pm 0.0003$ | $0.161 \pm 0.001$ |
| OMatG-TrigODE$\gamma$ | $76.0 \pm 0.2\,\%$ | $0.0529 \pm 0.0004$ | $0.160 \pm 0.001$ |
| OMatG-EncDecODE | $74.9 \pm 0.2\,\%$ | $0.058 \pm 0.001$ | $0.169 \pm 0.001$ |
| OMatG-SBDODE | $76.7 \pm 0.5\,\%$ | $0.061 \pm 0.001$ | $0.163 \pm 0.001$ |

Table 9: Standard errors for three generation runs from the same checkpoints reported for METRe, RMSE, and cRMSE values for the *carbon*-24-unique-$N$-split datasets.

| Method | *carbon*-24-unique-$N$-split (low $\rightarrow$ high) | | | *carbon*-24-unique-$N$-split (high $\rightarrow$ low) | | |
|---|---|---|---|---|---|---|
| | METRe (%) $\uparrow$ | RMSE $\downarrow$ | cRMSE $\downarrow$ | METRe (%) $\uparrow$ | RMSE $\downarrow$ | cRMSE $\downarrow$ |
| DiffCSP | $96.7 \pm 0.1\,\%$ | $0.4257 \pm 0.0005$ | $0.428 \pm 0.001$ | $100.0 \pm 0.0\,\%$ | $0.083 \pm 0.005$ | $0.083 \pm 0.005$ |
| FlowMM | $97.8 \pm 0.2\,\%$ | $0.4044 \pm 0.0004$ | $0.4066 \pm 0.0003$ | $100.0 \pm 0.0\,\%$ | $0.0420 \pm 0.0004$ | $0.0420 \pm 0.0004$ |
| OMatG-LinearODE | $96.1 \pm 0.4\,\%$ | $0.3974 \pm 0.0005$ | $0.4014 \pm 0.0002$ | $100.0 \pm 0.0\,\%$ | $0.047 \pm 0.001$ | $0.047 \pm 0.001$ |

Table 10: Standard errors for 350 generation runs from the same checkpoints reported for METRe, RMSE, and cRMSE values for the *carbon-NXL* dataset. For breakdowns by number of atoms per unit cell $N$: there are 196 generation runs for $N = 6$, and 124 generation runs for $N = 8$.

| Model | All $N$ | | | $N = 6$ | | | $N = 8$ | | |
|---|---|---|---|---|---|---|---|---|---|
| | Std. Match | RMSE ↓ | cRMSE ↓ | Std. Match | RMSE ↓ | cRMSE ↓ | Std. Match | RMSE ↓ | cRMSE ↓ |
| CSPNet | 47.3 % | $0.008 \pm 0.001$ | $0.266 \pm 0.013$ | 60.0 % | $0.005 \pm 0.001$ | $0.203 \pm 0.017$ | 39.0 % | $0.013 \pm 0.003$ | $0.310 \pm 0.022$ |
| aug-CSPNet | 47.7 % | $0.006 \pm 0.001$ | $0.264 \pm 0.013$ | 69.2 % | $0.005 \pm 0.001$ | $0.157 \pm 0.016$ | 26.0 % | $0.010 \pm 0.004$ | $0.373 \pm 0.019$ |

Table 11: Standard errors for 479 generations from the same checkpoints reported for METRe, RMSE, and cRMSE values for the *carbon-X* dataset.

| Model | carbon-X | | |
|---|---|---|---|
| | Std. Match (%) ↑ | RMSE ↓ | cRMSE ↓ |
| CSPNet | 100.0 % | $0.0007 \pm 1 \times 10^{-5}$ | $0.0007 \pm 1 \times 10^{-5}$ |
| aug-CSPNet | 100.0 % | $0.0007 \pm 1 \times 10^{-5}$ | $0.0007 \pm 1 \times 10^{-5}$ |

Table 12: Standard errors for 479 generations with $N = 6$ from the same checkpoints reported for METRe, RMSE, and cRMSE values for a modified *carbon-X* dataset, in which six additional unit cells of the same crystal structure—but with $N = 12$ atoms—have been added during training. We note the surprising discrepancy added by the addition of these six structures during training and generation for only $N = 6$ atoms.

| Model | carbon-X-mod | | |
|---|---|---|---|
| | Std. Match (%) ↑ | RMSE ↓ | cRMSE ↓ |
| CSPNet | 100.0 % | $0.060 \pm 0.005$ | $0.060 \pm 0.006$ |
| aug-CSPNet | 100.0 % | $0.084 \pm 0.005$ | $0.084 \pm 0.006$ |

## J   Binary search algorithm for determining match-boundary

We address the question of distinctness using the following method: within the *carbon*-24 dataset, all structures are compared to one another using the `StructureMatcher` with variable tolerance. For the upper triangular of the $10\,153 \times 10\,153$ matrix of structure comparisons, we calculate the tolerance at which the structure pairs from each row and column transition from matching to non-matching (the match-boundary) for each of `stol`, `ltol`, and `angle_tol`. Matches can be rejected for one of two reasons: if the choice of `stol` is lower than the RMSE of the match, or if there is significant structural dissimilarity such that no tolerance is sufficiently large in order to be considered matching. We find the tolerance at the match-boundary using a binary search method, except in the case of `stol`, where the binary search method is not necessary since the output RMSE from `StructureMatcher` is itself the `stol` at the match-boundary (we validated this by computing the matrix with using `stol` binary search). For each varied tolerance, the other two are held constant at the standard settings used in benchmarking generative models.

Below we provide the binary search algorithm utilized to find the tolerance at the match-boundary for a given pair of structures. We utilized 16 CPUs over approximately 3 days in order to compute the match-boundary tolerance for `ltol` and `angle_tol` (for a total of $\approx 1150$ CPU hours per tolerance) and approximately 2 days for `stol` (for a total of $\approx 770$ CPU hours).

```python
import numpy as np
from pymatgen.analysis.structure_matcher import StructureMatcher

def binary_search(s1, s2, tol_to_test, thresh=1e-4):
    """ Returns value of tol_to_test at match boundary for PyMatGen
        Structure types s1 and s2"""
    # Set L (left boundary) to 0 for all three tolerances
    L = 0

    # Ensure tol_to_test is a string and assert that it be an allowed
        option
    tol_to_test = str(tol_to_test)
    assert tol_to_test in ["ltol", "stol", "atol"]

    # For stol, the output RMSE is the value at the match-boundary
    if tol_to_test == "stol":
        # set other two tolerances loosely
        ltol = 0.3
        angle_tol = 10.
        # set R to be loosest tolerance for stol
        R = 0.5

        sm = StructureMatcher(ltol=ltol, stol=R, angle_tol=angle_tol)
        res = sm.get_rms_dist(s1, s2)
        if res is None:
            # return R=0.5 if there is no match
            return R
        else:
            # return the RMSE if there is a match
            return res[0]

    # binary-search for ltol or atol
    if tol_to_test == "ltol":
        # set other two tolerances loosely
        stol = 0.5
        angle_tol = 10.
        # set R to be loosest tolerance for ltol
        R = 0.3

        sm = StructureMatcher(ltol=R, stol=stol, angle_tol=angle_tol)
        res = sm.get_rms_dist(s1, s2)
        if res is None:
            # return R=0.3 if no match on first try
            return R
```

```python
        # enter while loop if matched on first try
        while L < R:
            mid = (L + R) / 2
            # use new value of ltol at midpoint between L and R
            sm = StructureMatcher(ltol=mid, stol=stol, angle_tol=
                angle_tol)
            res = sm.get_rms_dist(s1, s2)
            # if R and L are close enough return R
            if np.abs(R-L) <= thresh:
                return R
            # if match, move R to be at midpoint
            elif res is not None:
                R = mid
            # if not matching, move L to be at midpoint
            elif res is None:
                L = mid

    # binary-search for atol
    if tol_to_test == "atol":
        # set other two tolerances loosely
        ltol = 0.3
        stol = 0.5
        # set R to be loosest tolerance for atol
        R = 10.

        sm = StructureMatcher(ltol=ltol, stol=stol, angle_tol=R)
        res = sm.get_rms_dist(s1, s2)
        if res is None:
            # return R=10. if no match on first try
            return R
        # enter while loop if matched on first try
        while L < R:
            mid = (L + R) / 2
            sm = StructureMatcher(ltol=ltol, stol=stol, angle_tol=mid)
            res = sm.get_rms_dist(s1, s2)
            # if R and L are close enough return R
            if np.abs(R-L) <= thresh:
                return R
            # if match, move R to be at midpoint
            elif res is not None:
                R = mid
            # if not matching, move L to be at midpoint
            elif res is None:
                L = mid
```

