# OpenReview forum: "All that structure matches does not glitter"
_NeurIPS.cc/2025/Datasets_and_Benchmarks_Track — NeurIPS 2025 Datasets and Benchmarks Track poster_

### Official Review · Reviewer_u8DU · 2025-07-01

**Rating:** 5
**Confidence:** 3

**Summary:**

This paper identifies critical flaws in widely used datasets and benchmarks for generative models of inorganic crystal structures. The authors demonstrate that:

1. **Dataset non-uniqueness**: The *carbon-24* dataset contains only about 40% unique structures due to unit cell representation duplicates (Sec. 3.1, Fig. 1c-d).
2. **Inadequate data splits**: Random splitting in *perov-5* separates polymorphs across train/validation/test sets (Sec. 3.2, Fig. 1a-b), creating ill-posed evaluation tasks.
3. **Metric limitations**: Traditional match rate fails to account for polymorphism and high tolerances permit false matches (Sec. 2.3, Fig. 1e).

To address these, the authors:

- Release **revised datasets** (*carbon-24-unique*, *perov-5-polymorph-split*) with duplicates removed and polymorph-aware splits.
- Propose **two new improved metrics**:
    - **METRe** (Sec. 4.1): Evaluates coverage of reference structures by generated samples, robust to polymorphism.
    - **cRMSE** (Sec. 4.2): Penalizes non-matches by integrating RMSE with stol tolerance.

        Benchmarks on generative models (DiffCSP, FlowMM, OMatG) confirm improved evaluation rigor.

**Dataset Code Accessibility:**

Yes

**Dataset Code Comments:**

The authors provide a public release of data on Huggingface platform with detailed documentation.

**Ethical Considerations:**

No, there are no or only very minor ethics concerns

**Final Justification:**

The author addressed my concerns, provided additional experiments on Pymatgen StructureMatcher’s parametric sensitivity, and explained the complexity issues I was worried about. So I raise my score to 5.

**Limitations Weaknesses:**

- Duplicate identification relies on **Pymatgen’s tolerances** (Sec. 3.1.1). Authors acknowledge this may miss edge cases, requiring "human intervention" for perfect accuracy. And METRe/cRMSE might inherit **Pymatgen** **StructureMatcher’s parametric sensitivity** (e.g., stol/angle_tol choices impact scores).
- METRe requires **O(N×M) comparisons** (generated vs. test structures), becoming prohibitive for large datasets (Appendix A).
- I am wondering if your method can be extended to a larger dataset. Previously, Google DeepMind released a large crystal dataset[1], but this dataset was simulated by computation. It is conceivable that there must be a lot of repeating structures and polymorphs in it. Current experiments are limited to carbon/perovskites. It would be very interesting if it could be extended to a larger dataset.
I know that the current crystal generation models are only trained and tested on datasets such as carbon-20 and preov-5, so this is just a personal question and will not affect my score for the paper.

[1]Scaling deep learning for materials discovery, https://www.nature.com/articles/s41586-023-06735-9

**Strengths Contributions:**

- **Problem significance**: Exposes pervasive but overlooked flaws in CSP benchmarks (Sec. 1–3), validated via rigorous analysis. And gives quantitative analysis and visualization (e.g., Fig. 2 showing duplicate distributions).
- **Actionable solutions**:
    - Datasets: *carbon-24-unique* (40% duplicates removed), *perov-5-polymorph-split* (grouped polymorphs), and specialized sets (*carbon-X/NXL*) for equivariance testing (Sec. 3.1–3.2).
    - Metrics: **METRe** resolves polymorphism issues by matching "everyone to reference" (Sec. 4.1, Fig. 3d-e); **cRMSE** combines match rate and structural fidelity (Eq. 1, Appendix A).
- **Empirical validation**:
    - Shows METRe boosts "match" counts by 3–4× vs. standard metrics on *carbon-24-unique* (Table 1).
    - perov-5-polymorph-split* forces models to generalize to unseen compositions (Table 1), revealing true extrapolation capability.
- **Resource contribution**: All datasets/code released openly (Appendices B-C), enabling community adoption.

---

> ### Author Rebuttal · Authors · 2025-07-31
>
> We thank the reviewer for their thoughtful comments and suggestions for our manuscript. We detail our response to each topic separately.
>
> ## Tolerance-Dependent Results (Carbon-Unique Dataset)
>
> We agree with the reviewer that one of the weaknesses of our approach for deduplicating carbon-24 is having “edge cases” that are difficult to check without human intervention. We can quantify the quality of our deduplication method by recomputing the plots in Fig. 2 (which detail both the match-boundary density by structure, as well as the fraction of unique structures, as a function of tolerance value, for all three tolerances) for the new carbon-24-unique dataset and **add these figures to the manuscript**.
>
> ## Tolerance-Dependent Results (METRe/cRMSE)
>
> We thank the reviewer for pointing out the inheritance of parametric sensitivity for the METRe and cRMSE metrics. While we agree that this is true, we believe that a sensitivity analysis would help address concerns about the usefulness of the metrics.
>
> We perform this analysis by varying all three tolerances (ltol, stol, and angle\_tol) used in StructureMatcher by \+/- 10% of their values. We report below the highest and lowest scores of METRe rate and cRMSE—from varying ltol and angle\_tol—as a function of stol, for each of the three values of stol used in our sensitivity analysis. In the manuscript, **we can add the detailed results as two-dimensional heatmap figures,** which are slices of our three-dimensional tolerance space.
>
> We expect that the analysis will vary as a function of the quality of the generated structures, so we include analysis for both the best and worst performing OMatG positional interpolants (the Linear and Trigonometric+$\\gamma$ interpolants, respectively). In our response to Reviewer qSbc in which we benchmark performance of new MP-20 splits with standard tolerances (of ltol=0.3, stol=0.5, and angle\_tol=10.0), we note that the best OMatG model has METRe rate \= 70.50% and cRMSE (at stol=0.5) \= 0.187, while the worst OMatG model has METRe rate \= 42.78% and cRMSE (at stol=0.5) \= 0.373.
>
> | **Model** | **cRMSE (stol=0.45) low/high** | **cRMSE (stol=0.5) low/high** | **cRMSE (stol=0.55) low/high** |
> |-------------|-----------|-------------|-----------|
> | OMatG-best (Linear positional interpolant)  | 0.172 / 0.174  |  0.186 / 0.189 | 0.199 / 0.203 |
> | OMatG-worst (Trigonometric+$\\gamma$ positional interpolant)  | 0.341 / 0.346 | 0.369 / 0.376 | 0.395 / 0.403 |
>
> | **Model** | **METRe rate (stol=0.45) low/high** | **METRe rate (stol=0.50) low/high** | **METRe rate (stol=0.55) low/high** |
> |-------------|-----------|-------------|-----------|
> | OMatG-best (Linear positional interpolant)  | 68.26% / 69.43%  |  69.75% / 71.16% | 71.61% / 73.14% |
> | OMatG-worst (Trigonometric+$\\gamma$ positional interpolant)  | 36.70% / 39.50% | 41.22% / 44.61% | 46.09% / 50.18% |
>
> Our sensitivity results show that for a given choice of stol, both the METRe rate and the cRMSE values change negligibly for the structures generated by the OMatG model with the best-performing interpolant, by about 1% change in metric value for a \+/-10% change in the ltol and angle\_tol values. However, the metrics change more substantially for lower-quality matches via generated structures from the worst-performing interpolant. Our analysis highlights that **METRe and cRMSE metrics are most sensitive to stol for high-quality structural matches**, while metric performance is **more sensitive to all three tolerances for low-quality structural matches**. All the results of this work choose stol=0.5 which is consistent with other published works in generative materials models.
>
> ## Complexity Analysis for METRe Benchmark
>
> We would like to elaborate on the O(N✕M) complexity of the METRe computation (N being the size of the generated dataset, M being the size of the reference set, where in our case N=M).
>
> * While there are N by M function calls—as seen in the sample code provided in Appendix A—if two structures do not have the same stoichiometry, the computation is immediately terminated, and the expensive structure matching algorithm in StructureMatcher is not performed.
> * In the limit of the carbon dataset where all compositions are the same, the computation is the most expensive (worst case) as all NxN pairs need to be compared with StructureMatcher. We note, however, that after deduplication the size of the dataset has been significantly reduced compared to the original.
> * In the limit of more standard datasets with compositional diversity, such as perov-5 and MP-20, the computational cost is more reasonable because the majority of comparisons will immediately terminate.
>
> ## Extension to Larger Datasets
>
> We thank the reviewer for pointing out the potential of including the dataset published by Merchant et al. in the “Scaling deep learning for materials discovery” paper. We have performed polymorph analysis of the two datasets published at the corresponding github, and both datasets (one containing 554054 structures and the other smaller dataset containing 139541 structures). We determined that there are no polymorphs present in either of these datasets.
>
> However, we agree with the reviewer that extension to larger datasets that have more compositional diversity would be relevant to this manuscript. We direct the reviewer to our response to Reviewer qSbc (see ‘Extension to Other Datasets’), in which we detail how the methods presented in our manuscript can be extended to larger materials datasets which include polymorphs. In that response, we quantify the polymorph propensity by computing the number of unique compositions as a fraction of dataset size. We see that popular datasets such as MP-20 (total of 45229 structures) have approximately 82% unique compositions, and Alex-MP-20 (total of 675204 structures) has around 80% unique compositions. Thus, we **create new polymorph-aware splits** for MP-20, MPTS-52, and Alex-MP-20 and will publish these alongside this manuscript. We already benchmarked performance on the new MP-20 splits for the DiffCSP, FlowMM, and OMatG generative models (see section ‘Polymorph-aware MP-20 splits’, also in our response to Reviewer qSbc) and **show performance improvement in METRe rate, RMSE, and cRMSE metrics**.

---

> > ### Comment · Reviewer_u8DU · 2025-08-05
> >
> > Thank you for your detailed reply and additional experiments. My concerns have been addressed. I hope the authors will include additional experiments and comments in the final version of the paper, and I very much look forward to subsequent work extending to large-scale datasets.  so I will raise my score to 5.

---

### Official Review · Reviewer_syR1 · 2025-07-01

**Rating:** 4
**Confidence:** 5

**Summary:**

The paper addresses the challenges in the evaluation of generative models for inorganic crystal structure prediction (CSP), focusing on dataset quality, data splitting strategies, and evaluation metrics. It identifies issues in widely-used datasets like carbon-24 and perov-5, such as duplicate structures and improper handling of polymorphs, which can skew model performance metrics. The authors propose solutions, including deduplicated versions of the carbon-24 dataset, a new polymorph-aware split for datasets, and two novel evaluation metrics (METRe and cRMSE) designed to better handle polymorphs. The paper evaluates three generative models (DiffCSP, FlowMM, and OMatG) on these revised datasets and provides open access to the datasets and code for reproducibility. The work emphasizes the importance of robust benchmarks for advancing CSP and offers practical fixes to improve dataset quality and evaluation rigor.

**Additional Feedback:**

The paper makes valuable contributions to crystal structure prediction (CSP) by addressing dataset quality and evaluation metrics, particularly through deduplicated carbon-24 datasets, polymorph-aware splits, and new metrics (METRe, cRMSE). These are timely and impactful for materials science and ML.
### Suggestions for Improvement
1.	Generalizability: Discuss whether duplicate detection and polymorph splits apply to other datasets (e.g., organic crystals) to broaden relevance.
2.	Enantiomorphs: Include preliminary evaluation on carbon-24-unique-with-enantiomorphs or justify its deferral more clearly.
3.	Accessibility: Add a brief CSP introduction for non-experts to appeal to NeurIPS’s broad audience.
4.	Minor typo: On line 171, “superimoposed” should read “superimposed”.
### Questions for Authors
1.	Can you provide quantitative results showing how duplicates in carbon-24 affect model performance?
2.	How do METRe and cRMSE outperform existing metrics, such as k-match rate?
3.	Are your methods applicable to non-crystalline or organic materials datasets?
4.	Why was enantiomorph evaluation deferred, and can you include initial results?

**Dataset Code Accessibility:**

Yes

**Dataset Code Comments:**

The paper provides revised datasets (e.g., carbon-24-unique, carbon-24-unique-with-enantiomorphs) and code for calculating METRe and cRMSE metrics, with links to these resources detailed in Appendices B and A. This ensures accessibility and reproducibility, as the datasets are curated from existing sources (e.g., carbon-24 from DFT searches) and deduplicated using Pymatgen’s StructureMatcher, while the code is implemented in Python and documented for metric computation.

**Ethical Comments:**

The paper adequately addresses ethical considerations, with clear statements on societal impact, proper attribution, and LLM usage. A more thorough discussion of dataset biases or downstream implications would strengthen this aspect.

**Ethical Considerations:**

No, there are no or only very minor ethics concerns

**Final Justification:**

The paper makes a valuable contribution by improving the rigor of CSP evaluation through deduplicated datasets, polymorph-aware splits, and new metrics (METRe, cRMSE). These fixes address well-known issues in widely used benchmarks and will likely benefit future generative modeling work in materials science. The rebuttal further clarified scope, generalizability to other datasets, and the rationale for handling enantiomorphs, while committing to expanded dataset releases and clearer CSP background for non-experts. Nonetheless, the contribution remains mainly in dataset curation and evaluation methodology, with some gaps such as deferred enantiomorph evaluation and limited demonstration beyond selected datasets. Overall, I view the work as timely and impactful for the community, but not beyond a borderline accept. I therefore maintain my score of 4.

**Limitations Weaknesses:**

1.	The paper’s scope is limited to specific datasets (carbon-24, perov-5, MP-20) and models. While these are standard in CSP, the generalizability of the findings to other materials datasets or generative tasks is not discussed.
2.	The use of Pymatgen’s StructureMatcher for duplicate detection, while effective, is not a new tool. The novelty lies in its application, but the method itself is standard.
3.	The handling of enantiomorphs (Section 3.1.2) is described, but the paper defers model evaluation on the carbon-24-unique-with-enantiomorphs dataset to future work. This leaves a gap in demonstrating the practical utility of this dataset variant.

**Strengths Contributions:**

1.	The paper tackles a critical yet underexplored issue in CSP: the quality and suitability of datasets and evaluation metrics. This is highly relevant to the materials science and machine learning communities, as generative models for CSP are gaining traction for materials discovery.
2.	By identifying that the carbon-24 dataset contains only ~40% unique structures, the paper highlights a significant flaw in a standard benchmark, which could mislead model evaluations. This finding has immediate implications for researchers relying on this dataset.
3.	The introduction of deduplicated datasets (carbon-24-unique, carbon-24-unique-with-enantiomorphs) and polymorph-aware data splits addresses a practical need for more robust benchmarks, potentially improving the reliability of future CSP research.
4.	The proposed METRe and cRMSE metrics are designed to handle polymorphs more effectively than existing metrics like match rate, offering a more nuanced evaluation of generative models. This could set a new standard for CSP evaluation.
5.	The paper’s focus on minimal, information-rich datasets for efficient model evaluation is valuable for reducing computational costs during model development, aligning with the need for scalable and practical ML solutions in materials science.

---

> ### Author Rebuttal · Authors · 2025-07-31
>
> We thank Reviewer syR1 for their ample feedback, which we address below.
>
> ## Broadening Scope of Paper
>
> As noted in our response to reviewer qSbc (see ‘Extension to Other Datasets’), **we extend the scope of this work** by also making polymorph-aware splits of MP-20, MPTS-52, and Alex-MP-20, which are all datasets with greater chemical and structural diversity. To summarize, **we will publish these new splits and benchmark on each new dataset**. Performance metrics for new MP-20 splits are provided in the response to reviewer qSbc.
>
> ## Novel Use of StructureMatcher
>
> Reviewer syR1 is correct that PyMatGen’s StructureMatcher in itself is not a new tool. Our duplicate detection pipeline, described in Section 3.1.1 of our manuscript, however, is novel. In addition, it is rather general and can easily be extended to other datasets. PyMatGen’s StructureMatcher attempts to match similar structures (within a certain tolerances) while our analysis pipeline attempts to match identical structures (within a far smaller tolerance).
>
> We would also like to note that we will submit a pull request to PyMatGen in order to generalize its StructureMatcher to view enantiomorphs not as matching structures. This will allow us to properly analyze performance on the carbon-24-unique-with-enantiomorphs dataset (see discussion in next section).
>
> ## Enantiomorphs
>
> We welcome the opportunity to clarify the handling of the enantiomorph-labeled dataset, carbon-24-unique-with-enantiomorphs, in our manuscript. Because enantiomorphs are a specific kind of polymorph—that is, they are the chiral pair of a given structure—they are technically a distinct structure. In the carbon-24-unique dataset, we remove these chiral pairs, while in the carbon-24-unique-with-enantiomorphs dataset we keep them (and label the pairs).
>
> Our reasoning for not including carbon-24-unique-with-enantiomorphs in our evaluation is as follows:
>
> * There are only 80 enantiomorph pairs (totaling 160 structures) in the enantiomorph-labeled version of carbon-24-unique dataset, and we would not expect the results to substantially change as they are a very small fraction of the dataset (the enantiomorph-labeled dataset contains 4330 structures).
> * For all noncentrosymmetric structures that do not already belong to a chiral pair—which we can identify using the spglib space group identification package—we can enumerate new chiral pairs by using an inversion operation through the center of the unit cell. Beyond being informative, it is unlikely that the inclusion of new polymorphs—which are only allowed for structures in a small fraction of space groups (22 groups out of 230 total)—will be significant.
> * PyMatGen allows for improper rotations in the process of finding mappings between two input structures. Improper rotations include inversion, and thus StructureMatcher is able to find a mapping between two chiral pairs through inverting one structure to make the other—even though chiral enantiomorph pairs are distinct crystal structures with different handedness. Improper rotations can be disallowed by changing the base code within StructureMatcher (we did so in order to identify the chiral pairs). As noted in the above section, we plan to submit a pull request to the PyMatGen StructureMatcher. However, as the tool currently stands, it does not distinguish between enantiomorph pairs.
>
> We do, however, agree with the reviewer that adding an evaluation would be useful for completeness. **We suggest adding the following evaluations to the manuscript**:
>
> * A random split of carbon-24-unique-with-enantiomorphs and benchmarking using METRe.
> * An “overfitting” dataset of only chiral pairs, benchmarked similarly to the other “overfitting” datasets such as carbon-X and carbon-NXL datasets in this study.
> * We can additionally quantify in both cases if the model is able to evenly produce structures of the opposite handedness.
>
> ## CSP Introduction
>
> We thank Reviewer syR1 for the suggestion to add a brief introduction for crystal structure prediction (CSP) for non-experts to appeal to NeurIPS’s broad audience. **In the revised manuscript, we will extend the discussion of CSP in the introduction and make sure that this discussion is approachable by non-expert readers.**
>
> In addition, as highlighted in our response to Reviewer MkGQ (see ‘Providing Relevant Background’), we will release a publicly available notebook serving as a primer for non-expert readers to crystal structure prediction, duplicates, and polymorphs.
>
> ## Carbon Dataset Comparison
>
> As outlined in our response to Reviewer YQUV (see ‘Comparison of Original Carbon-24 and Deduplicated Carbon-24’), we do not report comparisons between models trained on the original and deduplicated carbon-24 datasets because such comparisons would not yield meaningful or trustworthy conclusions. In brief, any quantitative comparison would be confounded by the flawed nature of the original dataset. As a result, carbon-24 in its original form is not a reliable testbed for evaluating model performance. Our focus is instead on enabling meaningful evaluation through corrected splits and metrics. **We will summarize and include the clarifications from our longer response to Reviewer YQUV in the main text.**
>
> ## k-Match Rate Comparison
>
> We thank the reviewer for asking about k-match rate, and would like to both reiterate the discussion we provide in the manuscript and add points of clarification.
>
> * The $k$-match rate differs from METRe insofar as for each reference structure, $k$ structures of the same composition are generated. Then, these generated structures are compared to the reference structure, and if one of them matches the reference structure, it is considered a match. While $k$-match rate is “with respect to reference,” like METRe, it does not “match everyone”.
> * One weakness of $k$-match rate is its dependence on $k$. If $k$ is arbitrarily large, the model is more likely to correctly predict a valid structure that matches the reference structure, which it needs to do only once.
> * A second weakness of $k$-match rate is that it assumes $k$ is greater than or equal to the number of possible polymorphs for a given composition. For example, in many papers that our manuscript cites (refs. \[8, 9, 12, 14, 28\]), $k$ is set to 20\. In the MP-20 dataset, the highest occurrence of a single compound (SiO2) is 38 times, and in the Alex-MP-20 dataset the highest occurrence of a single compound is 46 times. The need to tune $k$ to account for the number of different plausible polymorphs makes the $k$-match rate less ideal than METRe for datasets with polymorphs.
> * We can think of METRe as encompassing a $k$-match rate, where $k$ is tuned to the number of polymorphs for each composition. Unlike the $k$-match rate, however, fewer structures are generated in the METRe evaluation process, which is more computationally efficient.
>
>
> **We will include these clarifications in the revised version of the manuscript.**
>
> ## Applicability to Other Systems
>
> The applicability of our work to other systems in chemistry and materials science, such as non-periodic systems or organic crystals, is summarized below:
>
> * **Identification and removal of duplicate data**: Deduplication of our data relies on the redundancy of the unit cell representation. For other periodic materials that use unit cells—such as molecular crystals—duplicate data arising from different unit cell choices, like in the carbon-24 dataset, may be a concern and can be addressed using the method we describe in this manuscript.
> * **Polymorph handling with METRe**: METRe is designed for use with any reference dataset in which all structures are unique, and is the correct benchmark to use (as opposed to standard match rate) in datasets with multiple entries of the same composition.
>   * An example application to a different non-periodic system are metamorphic proteins such as Lymphotactin which can exist in different folded conformations. One is a monomeric three-stranded β-sheet, the other is a dimeric β-sheet (see Camilloni and Sutto, “Lymphotactin: how a protein can adopt two folds” (2009) J. Chem. Phys. 131, 245105). The algorithm underlying METRe could be used in the context of generative protein structure prediction—replacing StructureMatcher with a more relevant match metric for proteins—to check for the prediction of both folded conformations for this protein.
>   * An example application to a different periodic system are molecular crystals which are well-known for their propensity to form a variety of crystal structures from the same molecule building blocks. The “problem” of polymorphism in these systems has undergirded decades of work on crystal structure prediction, which is highly relevant for applications such as drug design and manufacturing.
> * **cRMSE**: This metric is applicable in any context that uses StructureMatcher, and could be extended to any periodic crystalline system (inorganic crystals, molecular crystals, etc.).
>
>
> **We will include this discussion in the revised version of the manuscript.**

---

> > ### Comment · Reviewer_syR1 · 2025-08-05
> >
> > Thank you for your response. Most of my concerns have been addressed, and I will maintain my current score.

---

### Official Review · Reviewer_YQUV · 2025-07-02

**Rating:** 4
**Confidence:** 3

**Summary:**

This work improves crystal generative models pipeline from various perspectives. First, for training set, it should contain
 diverse and unique crystal structures, while authors find that the widely-utilized carbon-24 dataset only contains ≈ 40% unique structures, with duplicates differing only by the choice of unit cell representation. So a deduplicated version is provided.  Second, materials datasets
 need to be split with care. So the author propose to split by atom number. Third, metric for generative models can be misleading. This work propose METRe and cRMSE, new model evaluation metrics that can correctly handle materials datasets with polymorphs.

**Dataset Code Accessibility:**

Partly

**Dataset Code Comments:**

No Code for Models or Reproducing numerical result.

**Ethical Considerations:**

No, there are no or only very minor ethics concerns

**Final Justification:**

The new metric and new dataset split method for the work can benefit the development of CSP models. However, we maintain that providing new dataset split and metrics only is not significant contribution. So I keep weakly positive score.

**Limitations Weaknesses:**

1. There is no new dataset introduced, only new split of old dataset.

2. There is no comparison between model trained on original carbon-24 and deduplicated carbon-24, making the benefit of deduplication less convincible.

**Strengths Contributions:**

1. The improvement is wide spread and comprehensive. As it fix the evaluation routine from different perspectives and provide a more proper way to benchmark crystal models.

  1.1 They propose a deduplicated version of carbon-24 dataset, leading to a more robust benchmark.

  1.2 They propose METRe and cRMSE metrics for polymorphs.

  1.3 They propose dataset split by atom, making evaluation closer to real-world setting.

2. Clear motivation and figure illustration for the new metric and split.

---

> ### Author Rebuttal · Authors · 2025-07-31
>
> We thank Reviewer YQUV for their feedback and address their concerns below.
>
> ## Split of Existing Datasets
>
> The reviewer is correct that our work does not introduce an entirely new dataset. The creation of new high-quality materials datasets typically requires extensive experimental campaigns or high-throughput DFT pipelines, which are not the purpose of this work. Instead, our goal is to correct and reorganize widely used existing datasets—some of which were generated over many years—for more rigorous and informative benchmarking of generative models.
>
> The datasets we build upon are widely used in the generative materials modeling community. However, polymorphs and duplicates were previously ignored—leading to incorrect learning tasks and misleading benchmarks. Duplicate structures in the carbon-24 dataset mean there is significant data leakage between splits. Similarly, grouping polymorphs of the same composition within data splits—in datasets with more than one composition—prevents data leakage across the same chemical compositions, and ensures that the generative models can learn meaningful chemistry for polymorphs. Removing duplicates also means that our model is exposed to a larger variety of structural prototypes as a fraction of the total dataset size for a given composition during training. The new METRe and cRMSE metrics ensure that the model is able to faithfully and accurately predict possible structures for a given chemistry near the convex hull.
>
> Beyond the corrections via deduplicating and resplitting, we also organize data in ways that allow us to probe generative model performance in informative ways. The carbon-24-unique-N-split datasets allow the systematic study of how generative models extrapolate beyond their training data. The carbon-X and carbon-NXL datasets serve as “overfitting” tests while explicitly considering how well the model can learn invariance to the choice of unit cell. The updated manuscript **will include a new figure** that explains these specialized datasets.
>
> **In summary, the data itself is not new, but the new datasets and/or accompanying splits are either corrected or more informative.**
>
> ## Comparison of Original Carbon-24 and Deduplicated Carbon-24
>
> We do not provide comparisons between models trained on the original carbon-24 dataset and the deduplicated carbon-24 datasets because these would not be meaningful. We summarize our argument for why this is the case below:
>
> * Comparisons between models trained on the original and the deduplicated datasets require choosing a comparison metric and a test set.
>   * Using METRe on the **deduplicated test set** will lead to biased results if a model is trained on the original dataset. This is because there will be data leakage between the training and test datasets.
>   * Using METRe on the **original dataset test set** (containing duplicates) is biased because METRe is not designed to work with datasets which contain duplicate structures. This is noted in the manuscript, and all duplicate-only datasets such as carbon-X or carbon-NXL are benchmarked with standard match rate. For instance, in the case that the test set only contains two distinct structures, but the first structure is replicated 99 times, whereas the second one only appears once, generative models may achieve a METRe score of 99% by just generating the first structure **only once**.
> * Comparisons of performance with standard match rate do not make sense in the context of many polymorphs, which is the case in both the original and deduplicated dataset.
>
> The novel METRe and cRMSE metrics are only meaningful in conjunction with deduplicated datasets. In our opinion, the original datasets including duplicates should only be utilized in contexts where the prevalence of duplicates (which indicates the size of their basin of attraction) is meaningful for the machine-learning task at hand. We argue here that for crystal structure prediction, the diversity of generated structures should be prioritized instead. **We will summarize and include these clarifications in the main text.**
>
> ## Reproduction
>
> We would like to emphasize that Appendix A provides pseudocode for the calculation of the novel METRe and cRMSE metrics, and that all evaluated generative models in the manuscript (DiffCSP, FlowMM, and OMatG) are open source (see Appendix C).
>
> We are aware that training costs can become prohibitive, especially during hyperparameter optimization with respect to the new cRMSE metric. Therefore, **we will release the relevant checkpoints upon publication of this manuscript**. Also, we will submit a pull request to **include the computation of the METRe and cRMSE metrics** in the OMatG framework.
>
> As noted in our response to reviewer MkGQ (see ‘Providing Relevant Background’), **we will also publish a notebook which will include relevant code for the process of de-duplication and polymorph identification** within a dataset.

---

### Official Review · Reviewer_qSbc · 2025-07-02

**Rating:** 5
**Confidence:** 3

**Summary:**

The paper identifies the existing issue in the existing practice for model evaluation. Besides fixing the issues in the existing dataset, the paper also provides new metrics for evalution: METRe and cRMSE, that can correctly handle materials datasets with polymorphs.

**Additional Feedback:**

No.

**Dataset Code Accessibility:**

Yes

**Dataset Code Comments:**

The code links and the corresponding licenses are clearly included in the submission.

**Ethical Comments:**

No.

**Ethical Considerations:**

No, there are no or only very minor ethics concerns

**Final Justification:**

As the authors have adequately addressed my concerns, I recommend accepting this paper.

**Limitations Weaknesses:**

The paper focus mainly on carbon-24 and perov-5 datasets, can the analysis be applied to other existing datasets? It will be better to illustrate the extension on other datasets and show the significance of the issue in the broader area.

**Strengths Contributions:**

1. The paper identifies the issues in the existing datasets. Datasets collected from scientific domains that require specialized expertise are often expensive to obtain. Therefore, it is crucial to thoroughly understand their characteristics and maximize their utility. This paper explores strategies for leveraging existing datasets more effectively to enhance model evaluation—a key aspect when working with scientific data. The current random split of the dataset overlooks underlying duplications and polymorphs, which may introduce bias into the evaluation process. All these issues need experts to fully revise them.
2. The paper proposed the solution for the issues in the datasets. The paper identified the issues in carbon-24 and perov-5, proposing new splits on them and illustrated how the new splits match the expectation. These new cured datasets may lead to better benchmarks and inspire deeper investigation of existing datasets.
3. New metrics are presented. METRe and cRMSE are proposed to address the issues of matching rate and RMSE. Based on the experiments, the effectiveness of the metrics has been demonstrated.

---

> ### Author Rebuttal · Authors · 2025-07-31
>
> We thank Reviewer qSbc for their questions and positive evaluation of our paper.
>
> ## Extension to Other Datasets
>
> **We recognize the benefit of extending a polymorph-aware approach to such broad-scope datasets such as MP-20, MPTS-52, and Alex-MP-20, and believe that this would increase the impact of this work.**
>
> We estimate the polymorph prevalence in each dataset by calculating the fraction of unique **reduced** compositions per dataset. Reduced compositions are akin to a material’s stoichiometry and tell us about the number of polymorphs (non-unique reduced compositions) in the datasets.
>
> * MP-20: 82% unique compositions (37217 / 45229\)
> * MPTS-52: 87% unique compositions (35350 / 40476\)
> * Alex-MP-20: 80% unique compositions (539489 / 675204\)
>
> We have performed new splits of all of the above datasets and benchmarked performance using METRe for a new polymorph-aware split of MP-20. Unlike for the resplitting of the perov-5 dataset, we carefully considered how the propensity for a given composition to exhibit polymorphism could depend on the N-arity of the material (the number of unique elements in the crystal). Therefore, when creating new splits for MP-20, MPTS-52, and Alex-MP-20, we grouped polymorphs together while **ensuring that the distribution of N-arity did not change between the combined dataset and each of the splits**.
>
> ### Original MP-20 Splits
>
> We refer to the supplementary material (Appendix G) where the METRe rate is benchmarked on the original MP-20 split for DiffCSP, FlowMM, and OMatG models. OMatG models were hyperparameter tuned to optimize for lower cRMSE, while DiffCSP and FlowMM models were trained using published hyperparameters. We reproduce below the DiffCSP and FlowMM results, which use the same hyperparameters as the DiffCSP and FlowMM models in the following section on the polymorph-aware MP-20 split.
>
> | **Model** | **METRe rate** | **RMSE** | **cRMSE (stol=0.5)** |
> |-------------|-----------|-------------|-----------|
> | DiffCSP      |  58.8% | 0.064 | 0.244 |
> | FlowMM      | 67.0% | 0.067 | 0.210 |
>
> We also reproduce below the metrics for OMatG models from Appendix G. These models use different hyperparameters compared to the OMatG models reported in the following section on the polymorph-aware MP-20 split. For both sets of OMatG models (original and polymorph-aware MP-20 splits), OMatG models were hyperparameter tuned to optimize for lower cRMSE.
>
> | **Model** | **METRe rate** | **RMSE** | **cRMSE (stol=0.5)** |
> |-------------|-----------|-------------|-----------|
> | OMatG (Linear positional interpolant)  | 66.0% | 0.058 | 0.208 |
> | OMatG (Linear positional interpolant with $\\gamma$) | 34.1% | 0.206 | 0.405 |
> | OMatG (Trigonometric positional interpolant) |  66.5% | 0.072 | 0.215 |
> | OMatG (Trigonometric positional interpolant with $\\gamma$) | 62.1% | 0.119 | 0.264 |
> | OMatG (Encoder-Decoder positional interpolant) |  57.6% | 0.098 | 0.272 |
> | OMatG (Score-Based Diffusion positional interpolant) |  56.6% | 0.107 | 0.277 |
>
> ### Polymorph-Aware MP-20 Splits
>
> We include below the benchmarking of our polymorph-aware MP-20 split. We note that only DiffCSP and FlowMM use the same hyperparameters as for the original MP-20 split. All OMatG models have been hyperparameter tuned to optimize for lower cRMSE.
>
> | **Model** | **METRe rate** | **RMSE** | **cRMSE (stol=0.5)** |
> |-------------|-----------|-------------|-----------|
> | DiffCSP      |  53.14%    | 0.084  | 0.279 |
> | FlowMM        |   65.18%   | 0.079 | 0.226 |
> | OMatG (Linear positional interpolant) | **70.50%** | **0.056** | **0.187** |
> | OMatG (Linear positional interpolant with $\\gamma$) | 57.24% | 0.119 | 0.282 |
> | OMatG (Trigonometric positional interpolant) | 67.59% | 0.071 | 0.210 |
> | OMatG (Trigonometric positional interpolant with $\\gamma$) | 42.78% | 0.204 | 0.373 |
> | OMatG (Encoder-Decoder positional interpolant) | 58.06% | 0.064 | 0.247 |
> | OMatG (Score-Based Diffusion positional interpolant) | 52.20% | 0.129 | 0.306 |
>
> Notably, a decrease in performance can be compared directly for DiffCSP and FlowMM models which use published hyperparameters tuned for the original MP-20 split. This is unsurprising, as the hyperparameters were optimized without consideration of polymorphs.
>
> For the OMatG models with different hyperparameters, each optimized for lower cRMSE, the majority of positional interpolants **saw improvement in performance for the polymorph-aware split in comparison to metrics for the original MP-20 split**. We also emphasize the **state-of-the-art results for METRe and RMSE resulting from training on our polymorph-aware MP-20 split** with the OMatG Linear positional interpolant.

---

### Official Review · Reviewer_MkGQ · 2025-07-06

**Rating:** 5
**Confidence:** 1

**Summary:**

The paper audits two popular crystal-generation benchmarks and finds hidden flaws—duplicate structures in carbon-24 and polymorphs split across train/test in perov-5—that make standard “match-rate” scores misleading. It releases cleaned datasets (deduped carbon-24 variants, polymorph-aware perov-5 split, and duplicate-only sets for overfitting tests) plus new metrics (METRe for polymorph coverage and cRMSE for fidelity). Benchmarks with DiffCSP, FlowMM and OMatG show that results shift dramatically under the corrected data and metrics, highlighting challenges with larger unit cells and structural diversity. All datasets and code are open-sourced to set a sturdier baseline for future materials-generative models.

**Additional Feedback:**

I checked the dataset on huggingface and it is easy-to-access.

**Dataset Code Accessibility:**

Yes

**Ethical Considerations:**

No, there are no or only very minor ethics concerns

**Limitations Weaknesses:**

--Physical fidelity is not quantified. The paper does not report how deduplication or polymorph grouping influences downstream properties such as DFT formation energy or phonon stability. A small follow-up study (e.g., average energy error before vs. after cleaning) would reassure domain scientists that structural edits preserve real-world validity.
--Domain background is light for non-materials readers. Concepts like enantiomorphs, polymorph prototypes, and StructureMatcher parameters are introduced briefly; newcomers may struggle to reproduce the workflow. Adding a primer paragraph or a code notebook that walks through one duplicate-detection example step-by-step would make the benchmark more approachable to ML audiences outside crystallography.

**Strengths Contributions:**

The work is both novel and action-oriented—it doesn’t just point out that duplicates and naive match-rates mislead progress, it delivers cleaned datasets, principled metrics, and convincing benchmarks to raise the bar for crystal-structure generation research.
--Reveals hidden flaws in standard benchmarks: Quantifies massive duplication in the carbon-24 dataset (only ≈40 % unique structures) and shows that the usual perov-5 split scatters polymorphs across train/val/test—issues never documented before.
--Provides concrete, open-sourced fixes: Releases deduplicated datasets, a polymorph-aware perov-5 split, and duplicate-only sets, plus two new metrics (METRe and cRMSE) that fairly evaluate coverage + accuracy when polymorphs exist.
--Demonstrates impact on model rankings: Re-benchmarking DiffCSP, FlowMM, and OMatG with the corrected data/metrics flips scores and highlights performance drops with larger unit cells, proving that past results were misleading.

---

> ### Author Rebuttal · Authors · 2025-07-31
>
> We thank Reviewer MkGQ for their positive assessment of our paper, and address their concerns with our comments below.
>
> ## Impact of Deduplication and Polymorphism
>
> We would like to clarify the impact of duplicate structures or polymorphs in materials datasets:
>
> * Polymorphs are crystals with the same composition but different structure, and therefore also typically different properties such as thermodynamic stability (energy), phonon stability, etc.
> * What we term “duplicate” structures are exactly the same crystal but with different unit cell choices. This only changes the representation and not the periodic crystal itself. Thus, no properties change between duplicate structures.
> * Structural edits are not made to any individual crystals in the new datasets. Structures are either removed entirely, or regrouped into new splits.
>   * In the carbon-24 dataset, duplicate structures are removed to create carbon-24-unique.
>   * In our new split of the perov-5 dataset, structures were reorganized in order to confine polymorphs of the same composition to the same split (test, train, or val).
>
> **We will summarize and add these clarifications to the main body of the manuscript.**
>
> Based on these points, we make the following observations:
>
> * Because properties are computed on a per-structure basis, deduplication of the carbon-24 dataset will change the distribution of properties for the dataset. **We will add a figure to the manuscript showing the statistics for the zero-point energy before and after deduplication of the dataset.** This is straightforward as the original carbon-24 dataset is published with an accompanying energy per structure. We expect to see a change in the distribution of energies from the original to the deduplicated version, as duplicate structures have identical energy.
> * Because there is no deduplication for the perov-5 dataset (only rearrangement), we do not expect any significant change in the statistics of energy distributions between the old and new dataset splits. We do not think that one has to extend such analysis to the perov-5 dataset, as energies are not readily available for each structure, and performing energy calculations with DFT is a significant computational barrier.
>
> ## Providing Relevant Background
>
> We appreciate the reviewer’s comment regarding domain knowledge introduction for non-experts, and **will implement their suggestion of a publicly available notebook as a primer** alongside the publication of this manuscript. The notebook will contain the following:
>
> * Very small toy datasets containing polymorphs (one created from the carbon dataset and one from MP-20).
> * Duplicate creation code, in order to show how duplicates of the same crystal but different unit cell choices can be generated from any given crystal structure.
> * Duplicate detection and removal code, using PyMatGen’s StructureMatcher as utilized in this work on the carbon-24 dataset, and application to duplicate structures.
> * Radial distribution function plotting for comparing the structural signatures of two crystals (necessary but not sufficient for determining duplicates).
> * Structure visualization pipeline using ase or PyMatGen, to demonstrate examples of duplicates with different unit cells, polymorphs, and enantiomers (a special kind of polymorph).
> * Polymorph identification code.
>
> **This notebook will provide a way for non-materials scientists to understand the specific challenges relevant to the materials domain, especially for crystal structure prediction.**

---

> > ### Comment · Reviewer_MkGQ · 2025-08-08
> >
> > I appreciate the clear rebuttal and the additional clarifications on deduplication vs. polymorphism and the planned educational notebook. The distinction between true duplicates (unit-cell choices of the same periodic crystal) and polymorphs (distinct structures with shared composition) is now explicit, and the dataset operations—deduplicating carbon-24 into carbon-24-unique and re-splitting perov-5 to confine polymorphs within splits—are reasonable.

---

### Note · Authors · 2025-08-16

The authors would like to use this opportunity to highlight the contributions and impact of this paper.

1. **Refining benchmarks for crystal structure prediction.** Setting correct benchmarks is of primary importance for optimizing models and judging model performance. Previously, benchmarks such as match rate were erroneously applied to crystal structure datasets with polymorphs. We introduce an adapted version of match rate called **METRe** (“**M**atch **E**veryone **T**o **Re**ference”) which accounts for polymorphs explicitly, as well as **cRMSE,** which combines match rate and quality of matches (RMSE) into a single metric.
2. **Publishing refined datasets and new dataset splits.** Our investigation uncovered two major issues. First, the existence of datasets with many crystal replicas with different choices of unit cell, and second, the incorrect handling of polymorphs by random splitting of datasets for training. We publish several corrected datasets and dataset splits based on the carbon-24 and perov-5 datasets, and in the updated version of the paper we will include polymorph-aware splits for MP-20, MPTS-52, and Alex-MP-20.
3. **Providing additional datasets for overfitting tests**. We provide new test datasets cultivated from the original carbon-24 dataset to examine the impact of identical structures but with different unit cells, as well as two versions of the deduplicated carbon-24 split by increasing or decreasing unit cell size N. By benchmarking these datasets, we show a clear, isolated dependence of model performance on unit cell size and set the bar for future models to clear.
4. **Providing new evaluations with our methods and datasets**. We benchmark all new datasets with METRe (or standard match rate for overfitting tests) and cRMSE. In relevant cases, we show significant improvements in performance by properly handling polymorphs.

The work we present in this manuscript **will advance the field of crystal structure prediction for generative machine learning models**. Crystal structure prediction has a long history prior to its introduction in the machine learning context—within which this paper critiques widely utilized datasets and benchmarks that were ill-formulated for materials structure prediction. We propose rigorous solutions and more meaningful benchmarks which result in improved models. Ultimately, this work narrows **the gap between scientific domain knowledge and generative model benchmarks and evaluation**.

---

### Decision · Program_Chairs · 2025-09-18

**Decision:**

Accept (poster)

**Comment:**

The paper proposes novel splits and metrics for popular datasets for evaluating crystal generation, an important topic in AI for science. In that, the paper does not provide novel datasets in itself, but rather a detailed analysis of the duplication issues inherent in existing benchmarks. It also shows that the new splits change the ranking of existing models and analyses effects of data splits in detail.

All five reviewers provided comprehensive reviews and participated in the rebuttal as needed. The main original criticism was the consideration of only two (though standard) datasets in the submission. However, the authors provided similar results on existing datasets in the rebuttal.

Altogether, all reviewers recognize the importance of the paper's findings and recommend acceptance. Two of the five reviewers gave a borderline accept rating, mainly due to the fact that the contribution is limited in that it isn't a new dataset. However, they still acknowledge the work as "timely and impactful", and the work is in scope according to the submission guidelines.